# ACON: OPTIMIZING CONTEXT COMPRESSION FOR LONG-HORIZON LLM AGENTS

## ABSTRACT

Large language models (LLMs) are increasingly deployed as agents in dynamic, real-world environments, where success requires both reasoning and effective tool use. A central challenge for agentic tasks is the growing context length, as agents must accumulate long histories of actions and observations. This expansion raises memory costs and reduces token efficiency in long-horizon tasks, yet prior work on context compression has mostly focused on single-step tasks or narrow applications. We introduce **Agent Context Optimization** (ACON), a unified framework that optimally compresses both environment observations and interaction histories into concise yet informative condensations. ACON leverages compression guideline optimization in natural language space: given paired trajectories where full context succeeds but compressed context fails, capable LLMs analyze the causes of failure, and the compression guideline is updated accordingly. Furthermore, we propose distilling the optimized LLM compressor into smaller models to reduce the overhead of the additional module. Experiments on AppWorld, OfficeBench, and Multi-objective QA show that ACON reduces memory usage by 26-54% (peak tokens) while largely preserving task performance, preserves over 95% of accuracy when distilled into smaller compressors, and enhances smaller LMs as long-horizon agents with up to 46% performance improvement.

## 1 INTRODUCTION

Large language models (LLMs) have become the backbone of AI agents across diverse real-world tasks, leveraging language knowledge and reasoning to plan, act, and adapt in dynamic environments (Yao et al., 2023). These tasks often unfold over extended horizons: the agent must gather information, invoke tools, and revise plans based on feedback. In such settings, context is not auxiliary but foundational, as tool APIs and data formats lack standardization (Anthropic, 2024a), and even personalized environments differ in file structures or account data. Agents must therefore accumulate and maintain records of prior actions, observations, and world state. Losing this information, such as an email identifier, file version, or API format, can derail task success.

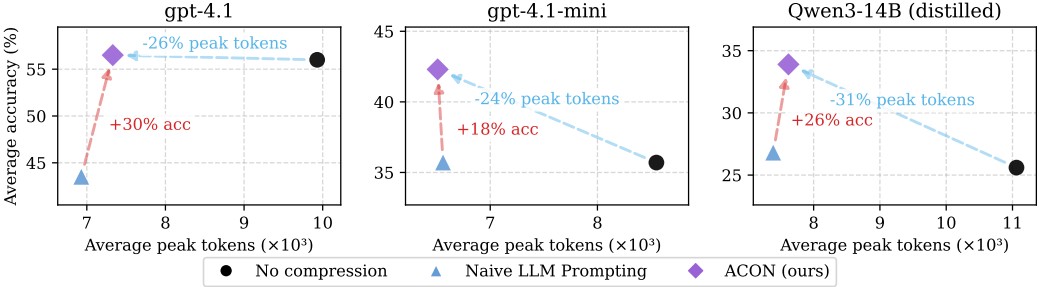

Figure 1: **Accuracy–Peak tokens trade-off** on AppWorld (Trivedi et al., 2024). We compare average accuracy versus peak input tokens in history compression. ACON (ours) reduces cost while preserving accuracy for the large model (gpt-4.1) relative to a naive prompting baseline, and even improves accuracy on smaller models (gpt-4.1-mini and Qwen-14B). More results are in Section 4.

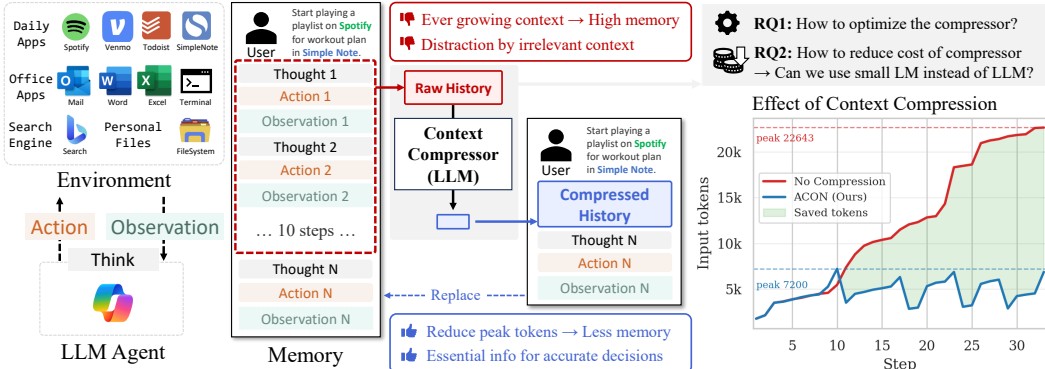

Figure 2: **Motivation: Unbounded context in LLM agents.** As LLM agents interact with environments, actions and observations continuously accumulate, leading to ever-growing contexts that incur high memory usage as in the red line on the right plot. This motivates *Agent Context Optimization* (ACON), which optimally compresses histories and observations into concise summaries, reducing peak tokens and memory as in the blue line on the right plot.

The centrality of context, however, makes it a bottleneck. As interactions accumulate, contexts grow unbounded as in Figure 2, creating two major challenges. First, the inference memory cost of transformers scales with the context length, which becomes prohibitive in long-horizon tasks. Second, excessively long contexts dilute relevant information, distracting the model with outdated or extraneous details (Shi et al., 2023). These issues make effective context management and compression indispensable. Existing compression approaches only partially address this need. Dialogue-oriented systems rely on session-level summarization or tiered memories suitable for conversational coherence but inadequate for multi-step carry-over (Packer et al., 2023). Document-centric methods in long-context QA or in-context learning (Li et al., 2023; Jiang et al., 2024) assume single-step reasoning where context can be discarded after producing an answer. While effective in their domains, these strategies fall short for complex multi-step tasks where success depends on retaining structured signals across many interactions.

The challenge becomes particularly critical in productivity scenarios (e.g., email management, document processing, or workflow automation) where agents must coordinate across heterogeneous tools and maintain precise state information. Unlike simpler agent tasks, these environments demand preservation of diverse signal types: factual history, action–outcome relationships, evolving environment states, success preconditions, and future decision cues. Naive strategies like token truncation or generic summarization easily lose critical details essential for long-horizon reasoning. Recent agent-focused compression methods (Deng et al., 2023; Lee et al., 2025; Smith, 2025) either specialize narrowly or rely on brittle heuristics, limiting their applicability across the full spectrum of multi-step agent tasks.

We address these challenges with **Agent Context Optimization (ACON)**, a unified framework for systematic and adaptive context compression. Our study yields three key findings. First, task- and environment-specific guidelines enable consistent context compression across diverse agents without sacrificing performance. Second, optimized contexts not only reduce memory costs but also improve decision quality, allowing smaller LLMs to act more effectively. Third, high-quality compressors can be distilled into even smaller models, reducing overhead and improving deployability.

ACON dynamically condenses environment observations and interaction histories into concise, informative representations. Rather than handcrafting prompts, we introduce a guideline optimization pipeline that refines compressor prompts via failure analysis in natural language space (Pryzant et al., 2023; Khattab et al., 2024; Yüksekgönül et al., 2025; Han et al., 2025), ensuring that critical environment-specific and task-relevant information is retained. Importantly, ACON is gradient-free, requiring no parameter updates, making it directly usable with closed-source or production models. We further distill optimized compressors into smaller models for cost-efficient deployment.

We validate ACON on three multi-step agent benchmarks: AppWorld (Trivedi et al., 2024), OfficeBench (Wang et al., 2024b), and Multi-objective QA (Kwiatkowski et al., 2019; Zhou et al., 2025), each requiring 15+ interaction steps. Our empirical results demonstrate clear advantages of ACON: (i) lowers memory usage by 26–54% (peak tokens) while largely maintaining task perfor-

mance; (ii) enables effective distillation of the context compressor into smaller models, preserving 95% of the teacher's accuracy across all benchmarks, thereby reducing the overhead of the additional module; (iii) allows small LMs to function more effectively as agents, improving performance by 32% on AppWorld, 20% on OfficeBench, and 46% on Multi-objective QA by mitigating the distraction of long contexts through ACON. Our result highlights on AppWorld benchmark are in Figure 1.

To summarize, our work makes the following contributions:

- We propose **Agent Context Optimization** (ACON), a framework for compressing both environment observations and interaction histories, tailored to multi-step, long-horizon agentic tasks.
- We develop a failure-driven, task-aware compression guideline optimization. Our approach is entirely **gradient-free** and readily applicable to any LLMs, including API-based models.
- We provide a cost-efficient solution by distilling optimized compressors into smaller models, preserving over **95%** of the teacher's performance while reducing module overhead.
- We validate ACON on AppWorld, OfficeBench, and Multi-objective QA, showing that it reduces memory usage by **26-54%** (peak tokens) while preserving task success with large LLMs, and enabling small LMs to achieve **20-46%** performance improvements.

## 2 RELATED WORKS

**Long-horizon LLM agents.** Large language model (LLM) agents extend pretrained models beyond static single-step reasoning tasks (e.g., RAG-based QA, math problem solving, or code generation) to interactive decision-making in dynamic environments (Yao et al., 2023; Wang et al., 2024a; Team et al., 2025; OpenAI, 2025a). Unlike chatbots or solvers that return an answer in one pass, agents must iteratively observe their surroundings, select tools, and execute actions while revising their plans based on feedback (Shridhar et al., 2021; Jimenez et al., 2024; Zhou et al., 2024; Wei et al., 2025; Xie et al., 2024; Bonatti et al., 2024). Recent work highlights the importance of *long-horizon LLM agents*, which tackle tasks that unfold over dozens to hundreds of steps and require coordination across multiple applications and tools (Kwa et al., 2025; Trivedi et al., 2024; Wang et al., 2024b; 2025b). A central challenge in these scenarios lies in managing the *dynamic long context*, where the agent must retain multi-step interaction histories and handle diverse observations produced by heterogeneous environments.

**Context compression for LLMs.** Managing this ever-growing context has been a longstanding challenge, and a variety of approaches have been proposed to compress LLM inputs. Prior works on context compression can be broadly grouped into three directions: document- or retrieval-based compression (Seo et al., 2025; Li et al., 2023; Xu et al., 2024; Yoon et al., 2024; Zhou et al., 2025; Jiang et al., 2024; Shandilya et al., 2025), dialogue memory summarization (Xu et al., 2025; Maharana et al., 2024; Wang et al., 2025a), and low-level KV cache compression (Zhang et al., 2025). While each line of research has demonstrated benefits in its respective setting, they remain insufficient for the dynamic and heterogeneous contexts required by long-horizon agents.

Beyond these directions, a few recent studies have attempted to compress context specifically for LLM agents (Deng et al., 2023; Lee et al., 2025; Smith, 2025; Yang et al., 2024b). However, these approaches either rely on naive prompting or target narrow domains such as the accessibility tree in web browsing, limiting their generality. In contrast, our work introduces a universal *agent context optimization* framework that is applicable to arbitrary agents. It supports both history and observation compression, and incorporates a generalizable optimization methodology that is agnostic to the underlying model, making it applicable to both open-source and proprietary API-based LLMs.

A closely related line of work is MEM1 (Zhou et al., 2025), which learns a compression policy jointly with the agent via reinforcement learning. This couples reasoning and compression and restricts applicability to models with trainable weights, whereas our method can work without weight updates. A more detailed comparison is provided in Appendix C.

## 3 AGENT CONTEXT OPTIMIZATION (ACON)

We present Agent Context Optimization (ACON), a unified framework for optimized history and observation compression in long-horizon LLM agents. We begin by formulating the agent task and

defining context cost in Section 3.1. Next, in Section 3.2, we introduce generative compression with LLMs for both history and observation, and formalize the associated optimization objective and its challenges. We then propose our optimization method in Section 3.3, followed by a distillation that enables smaller models for compressions to improve cost efficiency (Section 3.4).

## 3.1 PROBLEM FORMULATION

**Task.** An agentic task is formulated as a Partially Observable Markov Decision Process (POMDP) $\mathcal{E} = \langle \mathcal{S}, \mathcal{A}, \mathcal{O}, \mathcal{T}, \mathcal{R} \rangle$ with state space $\mathcal{S}$, action space $\mathcal{A}$, and observation space $\mathcal{O}$. The transition function $\mathcal{T}(s, a) \rightarrow (s', o)$ is deterministic, and it is determined by the implementation of the environment. Specifically, it executes an action $a \in \mathcal{A}$ in the environment and returns the next state and observation. The reward function $\mathcal{R}$ returns the reward given the terminal state $s_T$. The terminal state is arrived when the transition function receives the special action (e.g., `finish_task`).

An LLM agent interacts with the environment to get information for making a decision to achieve a given task $o_0$ through multiple steps. For each step $t$, the LLM $\mathcal{M}$ generates the action $a_t$ followed by its reasoning at each step (Yao et al., 2023; Wang et al., 2024a) given the interaction history $\boldsymbol{h}_{t-1} = (o_0, a_0, o_1, a_1, \cdots, o_{t-1}, a_{t-1})$ and the latest observation $o_t$:

$$\mathcal{M}(o_t, \boldsymbol{h}_{t-1}; \theta, \mathcal{P}_{\mathsf{agent}}) \mapsto a_t, \tag{1}$$

where $\theta$ refers to the pre-trained parameters of the LLM and $\mathcal{P}_{\mathsf{agent}}$ is the prompt that consists of a general environment description, tools, output format, and few-shot examples in natural language.

**Cost function for context.** We assume that the LLM agent's parameters $\theta$ and the task and system prompt $\mathcal{P}_{\mathsf{agent}}$ are fixed. We define a cost function $\mathcal{C}$ that measures the cost of encoding the dynamic context during action generation at each step such as $\mathcal{O}(n)$ computational cost of a transformer for decoding given $n$ input tokens. The cost function takes the interaction history $\boldsymbol{h}_{t-1}$, and the latest observation $o_t$ as input and returns the per-step cost:

$$C(\boldsymbol{H}) = \sum\nolimits_{t=1}^{T} \mathcal{C}(\boldsymbol{h}_{t-1}, o_t), \tag{2}$$

where $C$ is the total cost of completing the task, $\boldsymbol{H} = \{\boldsymbol{h}_{t-1}, o_t\}_{t=1}^{T}$ denotes the sequence of history and observation of each step. Typically, $C$ is proportional to the summation of token lengths of action and observations in each step, $\boldsymbol{h}_{t-1}$ and $o_t$. While the prompt cost is static and can be budgeted in advance, the costs from interaction histories are **unbounded**, leaving the user with only two options: terminate the task early or truncate the context heuristically to a maximum length. This raises the central question: *how can we compress context more effectively than such heuristics?*

## 3.2 HISTORY & OBSERVATION COMPRESSION WITH LLMS

To address this challenge, we use an LLM $f(\cdot; \phi, \mathcal{P})$, parameterized by pre-trained weights $\phi$ and a compression guideline $\mathcal{P}$, to minimize context cost defined in Equation 2 (e.g., *summarize the given interaction history*). As in Equation 1, the LLM receives two inputs at each step: the interaction history $\boldsymbol{h}_{t-1}$ and the latest observation $o_t$. This introduces two options for context compression:

**History compression.** The interaction history accumulates both environment observations and agent actions. In long-horizon tasks, this history can grow excessively large. To manage its length, we apply history compression only when the history length exceeds a predefined threshold $T_{\mathsf{hist}}$:

$$\boldsymbol{h}'_t = f(\boldsymbol{h}_t; \phi, \mathcal{P}_{\mathsf{hist}}) \text{ if } |\boldsymbol{h}_t| > T_{\mathsf{hist}}, \quad \boldsymbol{h}_t \text{ otherwise.} \tag{3}$$

The compressed history $\boldsymbol{h}'_t$ replaces the raw history in Equation 1. This selective compression ensures that the overhead of invoking the compressor is incurred only when necessary (Smith, 2025).

**Latest observation compression.** Given an action $a$, the environment returns an observation $o$ according to the transition function $\mathcal{T}(s, a) \rightarrow (s', o)$. We similarly apply observation compression only when the observation length exceeds a threshold $T_{\mathsf{obs}}$:

$$o'_t = f(o_t, \boldsymbol{h}_{t-1}; \phi, \mathcal{P}_{\mathsf{obs}}) \text{ if } |o_t| > T_{\mathsf{obs}}, \quad o_t \text{ otherwise.} \tag{4}$$

Figure 3: **Compression Guideline Optimization.** Feedback is generated by contrasting successful trajectories (no compression) with failed ones (with compression). The collected feedback is then used by LLM to refine the compression guidelines.

This mechanism avoids unnecessary overhead when $o_t$ is already short, while still reducing redundant or distracting content in long observations (Xu et al., 2024; Deng et al., 2023; Lee et al., 2025). The compressed one $o'_t$ replaces the raw one in Equation 1 and is stored in the interaction history $\boldsymbol{h}$.

In both cases, the compressor LLM selects information to preserve based on its learned prior knowledge of importance. However, there is **no guarantee** that the salient details required for successful task completion are retained. The agent context effectively serves as a **world model of the environment**, encompassing diverse forms of information such as causal relations (*e.g.,* email leaves drafts), evolving states (*e.g.,* account balance), preconditions (*e.g.,* login required), and task-relevant decision cues (*e.g.,* due dates). Effective context compression must therefore accommodate this heterogeneous and dynamic nature of agent context, ensuring that the most critical signals are preserved for long-horizon reasoning and task success.

**Optimization objective.** We optimize the compressor parameters $\psi \triangleq (\phi, \mathcal{P})$ to maximize task reward while minimizing context cost. At each step $t$, the compressor produces either a compressed history $\boldsymbol{h}'_t = f_{\mathsf{hist}}(\boldsymbol{h}_t; \psi)$ or observation $o'_t = f_{\mathsf{obs}}(o_t, \boldsymbol{h}_{t-1}; \psi)$. Let the compressed context be

$$\boldsymbol{H}'(\psi) = \{\boldsymbol{h}'_{t-1}, o'_t\}_{t=1}^T, \qquad C(\boldsymbol{H}'(\psi)) = \sum_{t=1}^T \mathcal{C}(\boldsymbol{h}'_{t-1}, o'_t). \qquad (5)$$

With the agent $\mathcal{M}(\cdot; \theta, \mathcal{P}_{\mathsf{agent}})$ fixed, the environment induces a trajectory $\tau(\psi)$ and terminal state $s_T(\psi)$ when the agent conditions on $\boldsymbol{H}'(\psi)$. Our learning objective is

$$\max_{\psi} \underbrace{\mathbb{E}\big[\mathcal{R}\big(s_T(\psi)\big)\big]}_{\mathsf{maximize}} - \lambda \underbrace{\mathbb{E}\big[C\big(\boldsymbol{H}'(\psi)\big)\big]}_{\mathsf{minimize}}, \qquad \lambda \geq 0, \qquad (6)$$

where $\lambda$ is a multiplier and the expectations are over tasks.

**Challenges.** The optimization objective in Equation 6 is difficult to optimize in practice because there is no gold supervision for compression, the reward is sparse and only revealed at the end of the trajectory, and the context cost is defined over discrete quantities, which precludes direct gradient computation. While these properties naturally motivate reinforcement learning (RL) (Sutton & Barto, 2018), applying RL to this setting introduces additional obstacles: (1) updating the parameters $\phi$ of a LLM with RL can be computationally prohibitive, (2) environment roll-outs are extremely expensive since each reward requires multi-step executions of both agent and compressor LLMs, and (3) policy gradient estimates suffer from high variance because compression quality is only indirectly evaluated through eventual task success.

### 3.3 OPTIMIZING COMPRESSION GUIDELINES

To overcome these challenges, we propose to optimize **compression guidelines** $\mathcal{P}$ (natural language prompts) for context compression, rather than fine-tuning model parameters $\phi$. Trajectories under compressed contexts provide *dense signals* about the quality of compression. For example, if the agent fails with compressed context while succeeding without compression, this indicates that the compressed context may have lost crucial information. Such trajectory-level comparisons yield richer feedback than scalar rewards (e.g., binary task success).

We instantiate this idea as prompt optimization using an LLM as the optimizer, where the natural language prompt $\mathcal{P}$ is refined via feedback expressed in natural language (Yang et al., 2024a; Yük­sekgönül et al., 2025; Khattab et al., 2024). We introduce **compression guideline optimization** based on *contrastive task feedback*.

On the training set $\mathcal{D}_{\text{train}}$, we run the LLM agent both without and with context compression to obtain baseline context $\boldsymbol{H}$ and compressed context $\boldsymbol{H}'$. We collect tasks where the agent succeeds with $\boldsymbol{H}$ but fails with $\boldsymbol{H}'$, forming a contrastive subset $\mathcal{D}_{\text{cont}}$. For each task in $\mathcal{D}_{\text{cont}}$, we query a optimizer LLM with the context before and after compression to obtain natural language feedback:

$$\text{Feedback} = \text{LLM}(\text{Feedback Instruction}, \boldsymbol{H}, \boldsymbol{H}'). \tag{7}$$

This feedback serves as a natural language gradient (Yüksekgönül et al., 2025), indicating how the compression guideline $\mathcal{P}$ should be refined. We then aggregate feedback from multiple trajectories:

$$\mathcal{P}^{(1)} = \text{LLM}(\text{Update Instruction}, \mathcal{P}^{(0)}, \|_{i=1}^{n}\text{Feedback}_i), \tag{8}$$

where $\|$ is concatenation of feedbacks from each task, which corresponds to a batch optimization step in textual gradient descent (Yüksekgönül et al., 2025). We also generate multiple candidate prompts $\{\mathcal{P}_k^{(1)}\}$, evaluate them on $\mathcal{D}_{\text{cont}}$, and select the best-performing one. We refer this process as *utility maximization* step $\overline{\text{UT}}$ as it primarily maximizes the first term (task reward) of Equation 6.

However, optimizing only for reward may neglect the context cost (second term in Equation 6). To address this, motivated by alternating optimization, we perform a second iteration that conditions only on successful task with compressed context, asking the LLM to generate feedback about which information was actually used during execution. This refines $\mathcal{P}^{(1)} \rightarrow \mathcal{P}^{(2)}$, encouraging shorter yet sufficient contexts. We refer this additional process as *compression maximization* step $\overline{\text{CO}}$ as it minimizes the second term (context cost) of Equation 6.

We illustrate overall process in Figure 3. Algorithm 1 and prompts are in Appendix B.

### 3.4 DISTILLING CONTEXT COMPRESSION INTO SMALL MODELS

While compression guideline optimization enables effective compression, repeatedly invoking the large LLM for compression adds substantial overhead. To reduce this cost, we **distill the com­pressor into a smaller model**. The teacher with optimized guideline $\mathcal{P}^*$ (parameters $\phi_{\mathsf{T}}$) generates compressed outputs $\boldsymbol{y}$ from input $\boldsymbol{x}$, which supervise the student (parameters $\phi_{\mathsf{S}}$). We train the student with a cross-entropy objective (Kim & Rush, 2016) with input-output pair $(\boldsymbol{x}, \boldsymbol{y})$, where $(\boldsymbol{x}, \boldsymbol{y}) = (\boldsymbol{h}_t, \boldsymbol{h}'_t)$ for Equation 3 or $(\boldsymbol{x}, \boldsymbol{y}) = ((\boldsymbol{h}_{t-1}, o_t), o'_t)$ for Equation 4:

$$\min_{\phi_{\mathsf{S}}} \ \mathbb{E}_{(\boldsymbol{x}, \boldsymbol{y}) \sim \mathcal{D}_{\text{train}}^{+}} \left[ -\sum_{n=1}^{L_{\boldsymbol{y}}} \log f(\boldsymbol{y}_n \mid \boldsymbol{x}, \boldsymbol{y}_{<n}; \phi_{\mathsf{S}}, \mathcal{P}^*) \right], \tag{9}$$

where $\mathcal{D}_{\text{train}}^{+}$ denotes tasks where the teacher succeeds with compressed context.

Once trained, the student replaces the teacher during inference, decoupling decision making from compression. This two-stage pipeline, guideline optimization then distillation, achieves effective compression with a much smaller model ($|\phi_{\mathsf{T}}| \gg |\phi_{\mathsf{S}}|$):

$$f(\cdot; \phi_{\mathsf{T}}, \mathcal{P}) \xrightarrow{\text{prompt optimization}} f(\cdot; \phi_{\mathsf{T}}, \mathcal{P}^*) \xrightarrow{\text{distillation}} f(\cdot; \phi_{\mathsf{S}}, \mathcal{P}^*). \tag{10}$$

## 4 EXPERIMENTS

We evaluate ACON on three challenging benchmarks that require multi-step interactions across di­verse domains. Our experiments are designed to address the following key questions:

- How well does ACON improve token efficiency while preserving performance? (Section 4.2)
- Does distilling the compressor reduce its size while maintaining agent performance? (Section 4.3)
- Can ACON help small, distilled LM agents perform better under long contexts? (Section 4.4)

Table 1: Results across different difficulty levels on **Appworld** benchmark (test-normal). Each block reports accuracy (task goal completion score), steps, peak input tokens ($10^3$), and dependency ($10^6$) for agents. Best results in each column are highlighted in bold. Rows in blue background indicate the results from ours . ACON consistently improves accuracy while reducing peak tokens and dependency, with ACON UTCO achieving the best overall performance.

| Method | Average (168) | | | | Easy (57) | | | Medium (48) | | | Hard (63) | | |
|---|---|---|---|---|---|---|---|---|---|---|---|---|---|
| | Acc. ↑ | Steps ↓ | Peak ↓ | Dep.↓ | Acc. ↑ | Peak ↓ | Dep.↓ | Acc. ↑ | Peak ↓ | Dep.↓ | Acc. ↑ | Peak ↓ | Dep.↓ |
| **Agent: gpt-4.1 / Compressor: gpt-4.1** | | | | | | | | | | | | | |
| No compression | 56.0 | **16.14** | 9.93 | 5.96 | 80.7 | 7.57 | 2.98 | 47.9 | 10.10 | 5.36 | **39.7** | 11.95 | 9.11 |
| **History Compression** | | | | | | | | | | | | | |
| FIFO | 45.8 | 28.48 | 6.73 | 5.69 | 84.2 | 5.85 | 2.89 | 39.6 | 7.26 | 6.24 | 15.9 | **7.14** | 7.80 |
| Retrieval | 27.4 | 33.17 | 8.39 | 6.68 | 61.4 | 7.40 | 3.97 | 12.5 | 8.74 | 7.72 | 7.9 | 9.02 | 8.33 |
| LLMLingua | 39.3 | 24.42 | 7.50 | 6.37 | 66.7 | 6.38 | 3.04 | 37.5 | 8.04 | 7.39 | 15.9 | 8.09 | 8.59 |
| Prompting | 43.5 | 24.01 | 6.93 | 5.29 | 66.7 | 6.36 | 2.84 | 41.7 | 7.10 | 5.36 | 23.8 | 7.31 | 7.48 |
| ACON UT | 51.2 | 20.92 | 7.17 | 4.49 | 77.2 | 6.45 | 2.43 | 50.0 | 7.39 | 4.47 | 28.6 | 7.65 | **6.37** |
| ACON UTCO | **56.5** | 22.82 | 7.33 | 4.69 | **86.0** | 7.09 | 2.84 | **56.2** | 7.48 | 4.43 | 30.2 | 7.44 | 6.55 |
| **Observation Compression** | | | | | | | | | | | | | |
| LLMLingua | 32.1 | 18.16 | 8.17 | 6.01 | 54.4 | 5.78 | 2.33 | 29.2 | 8.24 | 5.23 | 14.3 | 10.29 | 9.92 |
| Prompting | 42.3 | 17.38 | **6.58** | **4.09** | 64.9 | **4.92** | **1.88** | 35.4 | **6.96** | **4.11** | 27.0 | 7.79 | 6.07 |
| ACON UT | 47.0 | 16.67 | 7.62 | 5.08 | 70.2 | 5.87 | 2.21 | 45.8 | 7.79 | 5.00 | 27.0 | 9.07 | 7.73 |
| ACON UTCO | 53.6 | 18.12 | 7.43 | 4.93 | 82.5 | 5.66 | 2.63 | 47.9 | 7.30 | 4.43 | 31.8 | 9.14 | 7.50 |

## 4.1 EXPERIMENTAL SETUP

**Benchmarks & Metrics.** We focus on long-horizon agentic task benchmarks that require $10+$ interaction steps on average: **(1) AppWorld** (Trivedi et al., 2024): Main benchmark with 9 simulated apps (e.g., Venmo, Spotify, SimpleNote) and ∼100 simulated users. Performance is measured by task completion score. **(2) OfficeBench** (Wang et al., 2024b): Productivity tasks across 6 apps (e.g., Word, Excel, Email), operating on simulated documents. Performance is measured by benchmark-defined accuracy functions. **(3) 8-objective QA** (Kwiatkowski et al., 2019; Zhou et al., 2025): QA benchmark where agents interact with a search tool to answer 8 questions and output a consolidated answer set. Performance is the average of Exact Match (EM) and F1 scores across 8 questions.

In addition to task-specific performance metrics, we report three token efficiency metrics following prior work (Zhang et al., 2025; Zhou et al., 2025): **(1) Steps:** The average number of interaction steps per task. **(2) Peak Tokens:** The maximum context length encountered across all steps. **(3) Dependency:** The cumulative dependency of each generated action on prior tokens, measuring how much generation relies on the context history. Full details are provided in the Appendix B.

**Baselines.** **(1) No Compression:** full uncompressed context. **(2) FIFO:** keep the most recent $k$ interactions, discarding earlier ones (Yang et al., 2024b). **(3) Retrieval:** select $k$ past interactions most similar to the current query via embedding search (Xu et al., 2025). **(4) LLMLingua:** extractive compression with an encoder-only LM (Jiang et al., 2023; Pan et al., 2024). **(5) Prompting:** naive baseline using a general compression instruction (Smith, 2025; Lee et al., 2025).

**Our Methods.** We evaluate two versions of ACON. **(1) ACON UT** utilizes an *optimized guideline* for context compression after utility maximization step. **(2) ACON UTCO** applies compression maximization step CO after utility maximization UT, aiming for shorter but informative compression.

## 4.2 OVERALL PERFORMANCE AND TOKEN EFFICIENCY ON LLMS

In Table 1 and Table 2, we first evaluate ACON using on gpt-4.1 (OpenAI, 2025b) for both agent and compressor, which already achieves strong results on three long-horizon benchmarks.

For history compression, as shown in Table 1, on AppWorld, ACON reduces peak tokens by over 25% while preserving the accuracy of the no compression upper bound, outperforming all baselines that suffer severe degradation on medium and hard tasks spanning longer steps. On OfficeBench (Table 2a), ACON lowers peak context size by nearly 30% while maintaining accuracy above 74%. On 8-objective QA (Table 2b), ACON even surpasses the no compression baseline in EM/F1 while reducing peak tokens and dependency by 54.5% and 61.5%, respectively. For observation compression, ACON consistently outperforms all baselines confirming that compression guideline optimization is effective for compressing not only history but also raw observations.

Table 2: Results on **OfficeBench** and **8-objective QA** benchmarks. We report performance metrics (acc/EM/F1) along with steps, peak input tokens ($10^3$), and dependency ($10^6$). Best values are in **bold**. Rows in blue are ours. ACON consistently improves accuracy/efficiency trade-offs.

(a) OfficeBench

| Method | Acc. ↑ | Steps ↓ | Peak ↓ | Dep. ↓ |
|---|---|---|---|---|
| **Agent:** gpt-4.1 / **Compressor:** gpt-4.1 | | | | |
| No Compression | **76.84** | 11.52 | 7.27 | 4.43 |
| **History Compression** | | | | |
| FIFO | 67.37 | 12.26 | **4.02** | 2.64 |
| Retrieval | 65.26 | 16.20 | 4.33 | 2.06 |
| LLMLingua | 70.53 | 10.89 | 4.65 | 1.85 |
| Prompting | 71.58 | **10.13** | 4.40 | **1.10** |
| ACON UT | 74.74 | 13.13 | 4.93 | 3.85 |
| ACON UTCO | 72.63 | 11.54 | 4.54 | 1.91 |
| **Observation Compression** | | | | |
| LLMLingua | 71.58 | 11.89 | 7.38 | 6.14 |
| Prompting | 55.79 | 12.24 | 6.44 | 2.68 |
| ACON UT | 73.68 | 10.83 | 6.55 | 3.85 |
| ACON UTCO | 72.63 | 10.28 | 6.17 | 2.88 |

(b) 8-objective QA

| Method | EM ↑ | F1 ↑ | Steps ↓ | Peak ↓ | Dep. ↓ |
|---|---|---|---|---|---|
| **Agent:** gpt-4.1 / **Compressor:** gpt-4.1 | | | | | |
| No compression | 0.366 | 0.488 | 15.78 | 10.35 | 3.32 |
| **History Compression** | | | | | |
| FIFO | 0.293 | 0.388 | 19.26 | 5.09 | 2.51 |
| Retrieval | 0.331 | 0.438 | 20.06 | 5.11 | 2.62 |
| LLMLingua | 0.363 | 0.481 | 17.68 | 5.68 | 2.24 |
| Prompting | **0.376** | 0.478 | 18.70 | 4.73 | 1.66 |
| ACON UT | 0.373 | **0.494** | 17.14 | 4.71 | 1.57 |
| ACON UTCO | 0.335 | 0.458 | 17.79 | **4.65** | **1.50** |
| **Observation Compression** | | | | | |
| LLMLingua | 0.320 | 0.414 | 14.23 | 5.16 | 1.35 |
| Prompting | 0.288 | 0.397 | **11.64** | **3.41** | **0.45** |
| ACON UT | 0.364 | 0.475 | 16.33 | 4.97 | 1.28 |
| ACON UTCO | 0.336 | 0.461 | 14.00 | 4.22 | 0.81 |

Figure 4: **Results of distilled compressors on history compression** with gpt-4.1 as the agent. Grey dotted lines denote performance using the gpt-4.1 teacher compressor. Student models (Qwen3-14B, Qwen3-8B, Phi-4) are distilled from gpt-4.1 compressor using the optimized compression guideline after UT step, and evaluated across all benchmarks. We also include gpt-4.1-mini without distillation, showing that even a small model can serve as an effective compressor without additional training.

Applying only the utility maximization step (UT) improves performance while reducing token cost across all benchmarks, whereas the compression maximization step (CO) further lowers token cost but may slightly hurt accuracy, except in AppWorld where it even yields additional gains.

## 4.3 COMPRESSOR DISTILLATION

We distill the compressor with optimized guidelines after UT step into smaller models such as Qwen3-14B, Qwen3-8B (Yang et al., 2025), and Phi-4 (Abdin et al., 2024) using LoRA (Hu et al., 2022). As shown in Figure 4, the distilled compressors retain over 95% of the performance of the gpt-4.1 teacher (indicated by the grey dotted line) while reducing computational overhead. We also observe that gpt-4.1-mini, even without any distillation, can serve as an effective lightweight compressor on OfficeBench and QA. This indicates that small models can reliably replace large LLM-based compressors when equipped with optimized guidelines. These results confirm that small models are sufficient for compression, enabling the expensive LLM to be only reserved for the agent.

## 4.4 ACON FOR DISTILLED SMALL AGENTS

We examine whether ACON also benefits smaller LLM agents, which are particularly vulnerable to long-horizon inefficiency. Without compression, models such as Qwen3-14B often fail on medium and hard tasks due to distracting context. As shown in Figure 5, ACON substantially improves their performance: on AppWorld, Qwen3-14B improves from 26.8% to 33.9%, and on 8-objective QA from 0.158 to 0.197 EM. These results demonstrate that ACON acts as an equalizer, enabling smaller agents with concise but informative contexts to approach the performance of larger models.

## 4.5 ANALYSIS

**Compression threshold: moderate value yields the best trade-off.** In Figure 6, we provide ablations on threshold for compression in Equation 3 and Equation 4. Results show that smaller

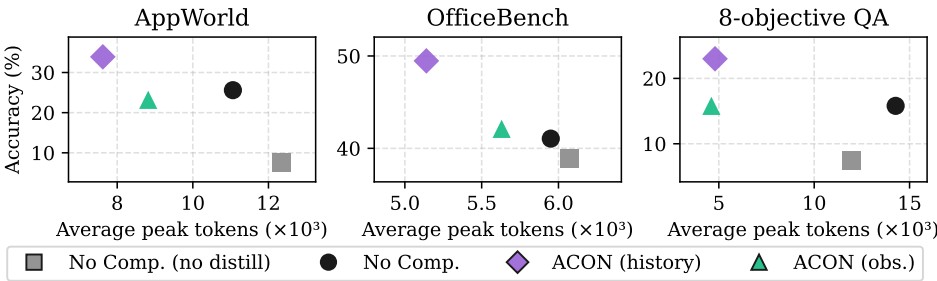

Figure 5: **Performance-efficiency trade-off of the Qwen3-14B agent** distilled from gpt-4.1 trajectories. For distilled compressors, we use the same distillation setting as in Figure 4. Compared to the baseline without compression, our framework ACON provides compressed trajectories combined with a distilled compressor, enabling the distilled agent to achieve consistently higher accuracy while requiring substantially fewer peak input tokens across all benchmarks.

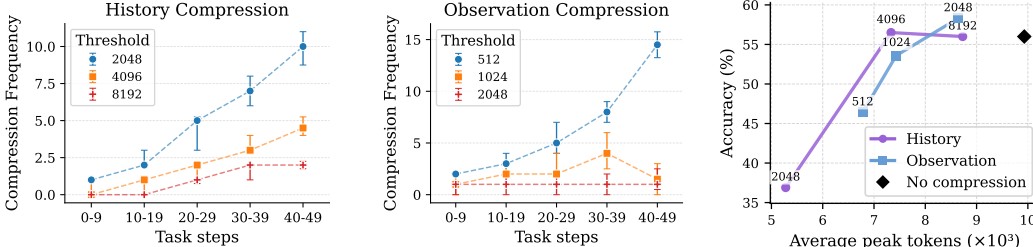

Figure 6: **Ablation studies on thresholds for compression** on AppWorld with gpt-4.1. (1) the number of compressions (compression frequency) for each length of task trajectories (task steps). (2) the performance comparison for each threshold setting.

thresholds reduce tokens but incur more frequent compression calls and degrade accuracy, while larger thresholds preserve accuracy with higher cost. Moderate values (4096 for history, 1024 for observation) provide the best trade-off, maintaining accuracy close to no compression while still reducing peak tokens substantially.

**Optimization cost.** It is important to clarify the cost of the guideline optimization introduced in Section 3.2, as it may initially appear expensive to practitioners. We compute all costs using the official API pricing described in Section B.2, and the results are summarized in Table 4. As shown, **the total cost is around \$20 for an AppWorld-scale domain.** The optimization consists of two phases: (1) collecting trajectories with the agent model and (2) optimizing the guideline with the optimizer model. For the rollout phase, we use fewer than 100 training examples for all benchmarks used (Appendix B.1).

Table 4: API cost for compression guideline optimization on AppWorld benchmark with 90 training examples.

| Stage (LM) | Cost (\$) |
|---|---|
| **Rollout** | |
| without comp. (gpt-4.1) | 10.7 |
| with comp. (gpt-4.1) | 8.9 |
| **Optimization** | |
| UT step (o3) | 1.2 |
| CO step (o3) | 0.7 |
| **Total** | **21.5** |

Executing both compressed and uncompressed trajectories with gpt-4.1 costs around \$20 for 90 examples. This cost can be amortized by integrating rollout collection into normal online task execution (Wang et al., 2025c), and it can be reduced further by using fewer examples when needed. Notably, we require at most two trajectories per example, which is substantially lower than reinforcement learning based approaches such as GRPO that require multiple trajectories per example for advantage estimation (Shao et al., 2024). For the optimization phase, we generate five candidate prompts per iteration and perform one UT step and one CO step. The total cost of this stage is under \$2 with the o3 model. Since our method does not require any model weight updates or GPU resources, it remains practical and easily applicable to both open and proprietary models via API access.

**Prompt optimizer: o3 + contrastive feedback works best.** We analyze how the choice of optimizer and the use of contrastive feedback affect compression guideline quality. As shown in Table 3, the default o3 with contrastive feedback yields the best performance, while removing contrastive feedback (only using failed trajectories) or switching to other models results in lower accuracy. Although o3 shows the best performance, we also demonstrate that the optimizer model can be re-

Table 3: **Ablation studies on the prompt optimizer** in AppWorld, gpt-4.1 agent and history compressor. Default is o3 optimizer with task contrastive feedback.

| Optimizer model | Task contrastive | Average Acc. |
|---|---|---|
| **o3** | ✓ | **51.2** |
| o3 | ✗ | 50.6 (-0.6) |
| gpt-4.1 | ✓ | 47.6 (-3.6) |
| gpt-5 | ✓ | 50.6 (-0.6) |

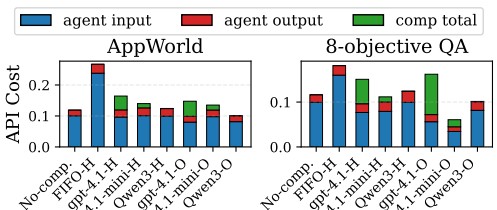

Figure 7: **API cost comparison.** H/O denote history/observation compression. Listed models are compressors; the agent is gpt-4.1.

placed to weaker models such as gpt-4.1, showing it still yields sufficiently fine guideline compared to the baseline guideline.

**Cost analysis with an API cost proxy.** An important point of discussion is the extent to which ACON improves efficiency. To clarify, ACON primarily targets *memory efficiency* (*e.g.,* reducing peak tokens and dependency), rather than providing overall API or computational cost reductions. Our experiments show that ACON reliably reduces memory cost while preserving or even improving task performance. Nonetheless, to prevent misunderstandings and to better inform future work, we also analyze whether ACON reduces end-to-end computational cost. We use API cost as a proxy for computational overhead, accounting for discounted cached tokens as described in Section B.2.

We report the API cost analysis in Figure 7, using gpt-4.1 as the agent and gpt-4.1, gpt-4.1-mini, and Qwen3-14B as compressors. The analysis yields three observations. First, *using small models reduces compressor cost*: replacing gpt-4.1 with smaller models such as gpt-4.1-mini or Qwen3-14B lowers overhead while preserving compression quality. Second, *observation compression reduces total API cost*: in AppWorld, using Qwen3-14B compressor lowers the per-

Table 5: Effect of compression threshold on peak tokens and API cost.

| Thr. | Peak ($10^3$) | Agent ($) | Comp. ($) |
|---|---|---|---|
| 2048 | 5.36 | .1440 | .0720 |
| 4096 | 7.33 | .1253 | .0360 |
| 8192 | 8.70 | .1179 | .0140 |

task cost by 15% (from $0.119 to $0.101) because observations are compressed before being cached in the agent. Third, *history compression may increase or decrease total cost depending on the task*. In AppWorld, cost increases by 4% (from $0.119 to $0.124) with Qwen3-14B compressor due to KV-cache invalidation and occasional increases in task length. In contrast, in 8-objective QA, cost decreases by 4% (from $0.116 to $0.112) with gpt-4.1-mini compressor, as fewer compression events lead to minimal KV-cache disruption while still shortening inputs. The benefit of history compression therefore depends on how often the compressed history is reused, which is controlled by the compression threshold. As shown in Figure 6 and Table 5, the threshold determines compression frequency and trades off peak tokens against total API cost. We recommend tuning it on a validation set to achieve the desired balance.

We include a more detailed discussion of the limitation, together with practical guidance on threshold selection and compressor choice, in Appendix A.

## 5 CONCLUSION

We presented **Agent Context Optimization** (ACON), a unified framework that systematically compresses both interaction histories and environment observations for long-horizon LLM agents. Unlike prior work that relies on naive prompting or narrow domains, ACON introduces compression guideline optimization in natural language space, enabling adaptive and model-agnostic compression. Experiments on AppWorld, OfficeBench, and Multi-objective QA show that ACON reduces peak tokens by 26-54% while maintaining or even improving task success. Beyond memory efficiency, we demonstrate that optimized compressors can be distilled into smaller models, substantially lowering overhead without sacrificing performance. Moreover, by supplying concise yet informative contexts, ACON allows small agents such as Qwen3-14B to approach the performance of much larger models. Overall, our findings highlight that ACON lays a foundation for more general, cost-effective, and deployable long-horizon LLM agents.

ETHICS STATEMENT

This work investigates optimized context compression framework for long-horizon LLM agents. It **does not** involve human subjects, user studies, or the collection of personally identifiable information. All experiments are conducted on **publicly available** benchmarks released under their respective licenses, which, to the best of our knowledge, do not contain sensitive personal data.

REPRODUCIBLE STATEMENTS

We conduct all experiments using the Azure OpenAI endpoint with fixed model snapshots and seed, specifically `gpt-4.1-2025-04-14` and `gpt-4.1-mini-2025-04-14`, to ensure reproducibility. Our implementation relies on PyTorch (Paszke et al., 2019), the Hugging Face Transformers library (Wolf et al., 2020), and the TRL library for training and vLLM (Kwon et al., 2023) for inference. Additional implementation details are provided in the Appendix B. We will also release our codebase to enable the research community to fully reproduce, verify, and extend our work on long-horizon LLM agents.

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

## A    LIMITATIONS & FUTURE WORKS

Our work addresses the context management problem in long-horizon LLM agents and proposes a framework for optimized context compression. While the method effectively reduces token costs with minimal performance degradation, it also presents several limitations.

A primary limitation is computational overhead. As discussed in Section 4.5, history compression can in some cases slightly increase total cost, since additional steps may be required to complete challenging tasks with the compressed history. Moreover, it breaks the KV-cache of transformer-based LLMs, which forces re-computation of compressed histories. This effect is particularly evident in the FIFO baseline. Observation compression alleviates some of this overhead, but the generative compression procedure itself introduces latency, slowing down agent response time (Lee et al., 2025). A promising future direction is the development of KV-cache–level compression or eviction strategies. Prior work has considered KV-cache compression for single-step reasoning (Zhang et al., 2025), long documents (Xiao et al., 2024; Willette et al., 2025), and long conversations (Kim et al., 2025). However, its role in multi-turn, long-horizon agents remains underexplored.

Another limitation is model and benchmark coverage. Our experiments primarily evaluate GPT models due to budgetary constraints. Although the framework is designed to be model-agnostic, its generalizability to other foundation models such as Gemini (Comanici et al., 2025) or Claude (Anthropic, 2024b) remains unverified. Similarly, we were unable to include large-scale open-source models such as DeepSeek-R1 (DeepSeek-AI et al., 2025) or Qwen3-235B (Yang et al., 2025) due to limited resources. Extending the analysis to these models would provide stronger evidence of robustness and broaden the applicability of our conclusions. We also validated our approach on three benchmarks. While these benchmarks reflect realistic agent settings, full real-world deployability remains an open question. Evaluating the method in real-world environments, where tasks are less controlled and constraints are more complex, would be a valuable direction for future work.

Furthermore, the optimization of the compression guideline also has limitations. Our method, along with prior prompt optimization approaches (Yüksekgönül et al., 2025; Khattab et al., 2024; Pryzant et al., 2023; Yang et al., 2024a), supports the view that updating natural language instructions using LLM-generated feedback is a valid strategy for improving LLM systems. However, unlike numerical gradient-based methods, this optimization provides no convergence guarantee. We partially address this issue by sampling multiple candidate guidelines and selecting one during training, but this remains a heuristic solution. A deeper analysis of the optimization process and more principled methods for optimizing the objective in Section 3.2 would be valuable directions for future work.

Another limitation concerns distillation quality. Although our distilled models retain most of the teacher's behavior, they do not achieve perfectly identical performance. We expect that this gap can be further reduced by increasing the amount of training data beyond the 100 examples per domain used in this work.

## B    EXPERIMENTAL SETUP DETAILS

### B.1    DATASETS

**AppWorld (Trivedi et al., 2024).**    AppWorld is our primary benchmark, providing a high-quality execution environment that integrates nine everyday applications (Spotify, SimpleNote, Amazon, Venmo, Gmail, Splitwise, File system, Todoist, and Phone) through 457 APIs. It also includes realistic simulations of approximately 100 functional users. This benchmark is particularly suitable for evaluating long-horizon productivity agents, as its multi-step tasks require an average of 42.5 API calls per task. We follow the official split, using 90 training tasks for guideline optimization and distillation, and 168 test-normal tasks for evaluation. An example trajectory from AppWorld is provided in Example E.1.

**OfficeBench (Wang et al., 2024b).**    OfficeBench is a benchmark for office automation using applications such as Word, Excel, PDF, Calendar, Email, Shell, and Calculator. It evaluates the ability of agents to coordinate across multiple apps to complete complex tasks, making it well suited for long-horizon scenarios. Tasks are categorized as 1-app, 2-app, or 3-app depending on the number of applications required. We restrict our experiments to text-related tasks, excluding those requiring

Table 6: Example tasks across benchmarks.

| Benchmark / Difficulty | Example Task |
|---|---|
| **AppWorld** | |
| Easy | Mark "Taking a solo backpacking trip" in my Bucket List Simple Note as not done. |
| Medium | Like all the Venmo transactions from today involving any of my roommates on my Venmo social feed. |
| Hard | Start playing a playlist on Spotify that has enough songs for my workout today. I do not want to have to change the playlist in the middle of my workout. The workout plan is in Simple Note. |
| **OfficeBench** | |
| 1-app | Create a new Word file called `random_paragraph.docx` and add the content in `random_paragraph.txt` to it. |
| 2-app | Analyze Excel data of students' grade and generate a teaching report in `teaching.docx`. |
| 3-app | Read company revenues and send an email with subject `revenues`, containing data to Bob for reporting, also write a `revenue.docx` to summarize it. |
| **8-objective QA** | |
| – | who wrote the song *Oceans Where Feet May Fail*?; who plays Eddie the Eagle in the movie?; when was the last time England were in the final of World Cup?; who plays Chelsea's mom on *Young and the Restless*?; what is the largest coin in the US?; who sang *Even the Bad Times Are Good*?; who sings *This Is My Town* country song?; which of the Guianas is not an independent country? |

OCR, as OCR quality could confound the evaluation. Since no official split is available, we randomly partition the tasks into training and test sets with a 1:1 ratio, resulting in 92 training tasks and 95 test tasks. We additionally refine the dataset by removing ambiguous tasks and ensuring that synthetic files (testbeds) are not shared across splits.

**8-Objective QA (Zhou et al., 2025).** The 8-objective QA benchmark simulates deep research-style agentic tasks. Unlike conventional multi-hop QA, which requires answering a single question using multiple pieces of evidence, this benchmark poses eight distinct questions within one task, and the agent must provide answers to all of them at the end. This design creates a more challenging setting for long-horizon agents. Following Zhou et al. (2025), we construct each task by grouping eight questions together. Questions are drawn from NaturalQuestions (Kwiatkowski et al., 2019), resulting in 100 training tasks (from the train split) and 100 test tasks (from the test split). For retrieval, we use a BM25 retriever over the 2018 Wikipedia knowledge base, following Jin et al. (2025).

We include the example task of each benchmark in Table 6.

### B.2 EVALUATION METRICS

For efficiency evaluation, we adopt two metrics—*peak tokens* and *dependency*—introduced in Light-Thinker (Zhang et al., 2025) and MEM1 (Zhou et al., 2025).

**Peak tokens.** Peak tokens are measured as the maximum number of tokens observed in any single sequence throughout the agent's trajectory, excluding system prompts. This metric serves as a proxy for inference-time memory requirements and corresponds to the maximum peak shown in Figure 2.

**Dependency.**    Dependency is defined as the area under the curve in Figure 2. At each step $t$, given the number of input tokens $n_i^{(t)}$ and output tokens $n_o^{(t)}$, it is calculated as:

$$\sum_{t \in [T]} \frac{(n_i^{(t)} + 2n_o^{(t)}) \times n_o^{(t)}}{2}. \tag{11}$$

This metric approximates the cumulative computational cost incurred by action generation across the trajectory.

**API Cost.**    For the cost analysis in Section 4.5, we use the official OpenAI pricing (as of September 2025) for `gpt-4.1` and `gpt-4.1-mini` (OpenAI, 2025b). Specifically, `gpt-4.1` is priced at \$3.00 per 1M input tokens, \$0.75 per 1M cached input tokens, and \$12.00 per 1M output tokens. For `gpt-4.1-mini`, the costs are \$0.80 per 1M input tokens, \$0.20 per 1M cached input tokens, and \$3.20 per 1M output tokens. For `Qwen3-14B` (Yang et al., 2025), since no official API pricing is available, we approximate the cost using OpenRouter[1] : \$0.06 per 1M input tokens, \$0.015 per 1M cached input tokens, and \$0.24 per 1M output tokens.

### B.3    Implementation Details & Hyperparameters

**API Inference.**    We set temperature 0.0 and fix the seed 42. Note that there is still non-determinism with fixing the seed and setting temperature as 0. To reduce the instability, we use the API snapshot form Azure OpenAI endpoint `gpt-4.1-2025-04-14` and `gpt-4.1-mini-2025-04-14`.

**Compression.**    For history compression, we set $T_{\text{hist}} = 4096$ for AppWorld and OfficeBench, and 2048 for 8-objective QA. We keep the last action, observation pair to preserve the latest information. This is the same for ACON and all baselines. For observation compression, we set $T_{\text{obs}} = 1024$ for AppWorld, 512 for OfficeBench, and 400 for 8-objective QA.

**Prompt Optimization.**    We use the OpenAI `o3` model (OpenAI, 2025d) for both analysis and update of prompts. During the update stage, we sample 5 candidate prompts and select the one that performs best on a subset of the training set.

The prompts used in each stage and step are provided as follows:

- Analysis prompt for $\overline{\text{UT}}$ step: Prompt E.1
- Update prompt for $\overline{\text{UT}}$ step: Prompt E.2
- Analysis prompt for $\overline{\text{CO}}$ step: Prompt E.3
- Update prompt for $\overline{\text{CO}}$ step: Prompt E.4

We also provide the detailed procedure in Algorithm 1. For the subset used in prompt selection during the $\overline{\text{UT}}$ step, we consider training tasks in $\mathcal{D}_{\text{cont}}^{(r)}$ where the agent succeeds without compression but fails with compression. For the $\overline{\text{CO}}$ step, we use training tasks in $\mathcal{D}_{\text{succ}}^{(r)}$ where the agent succeeds with compression. We perform one round consisting of a single $\overline{\text{UT}}$ step and a single $\overline{\text{CO}}$ step to obtain the guidelines used in our experiments, unless otherwise noted.

**Baselines**    For FIFO, we keep last 5 interaction turns which fits in similar compression rate in average with ACON. For retrieval, we also retrieve 4 interaction turns and keep the last turn. We use OpenAI text-embedding-3-large for embedding. For LLMLingua, we set keep rate as 30% for both history and observation. For naive prompting, we use the similar prompt with Lee et al. (2025) and do some human prompt engineering to specialize each prompt to history or observation compression.

**Compressor & Agent Distillation**    Both compressor and agent distillation use LoRA (Hu et al., 2022) with rank 16, $\alpha = 32$, learning rate $10^{-4}$, 3 epochs, batch size 4, and maximum sequence length 10,000. We adopt linear warmup (5% ratio), weight decay 0.01, and AdamW optimizer. No hyperparameter tuning was performed; the same setup is applied across all models and benchmarks.

---

[1]https://openrouter.ai/

We sample a single generation from the teacher for fine-tuning, leaving potential improvements from hyperparameter tuning or multi-sample training for future work. We use 1 A100 80GB GPU for both training and inference. For inference of fine-tuned models, we use greedy decoding (temperature 0.0).

## C ADDITIONAL RESULTS

We provide additional quantitative results to complement the main experiments in Section 4.

**OfficeBench difficulty breakdowns.** We further analyze OfficeBench with gpt-4.1 by difficulty level. The detailed breakdown in Table 8 shows that ACON yields the largest gains on the most challenging tasks in Level 3.

**Experiments with gpt-4.1-mini.** Results for the smaller variant gpt-4.1-mini (OpenAI, 2025b) across three benchmarks are reported in AppWorld (Table 9), OfficeBench (Table 10), and 8-objective QA (Table 11). The trends of ACON are consistent with those for gpt-4.1 in Section 4. In particular, Table 9 shows that history compression improves the performance of gpt-4.1-mini compared to the baseline, complementing the findings in Section 4.4 that ACON enhances the effectiveness of smaller LM agents. These results highlight the robustness of our method under resource-constrained settings.

**Experiments with gpt-5-chat.** We also evaluate on AppWorld using gpt-5-chat (OpenAI, 2025c), as reported in Table 12. The improvements follow the same trend as with gpt-4.1, demonstrating that ACON generalizes to the latest stronger proprietary models.

**Distilled optimizer.** Additional results for the distilled optimizer in AppWorld are shown in Table 13. Beyond the analysis in Section 4.3, we also include experiments where the compressor is distilled using guidelines without optimization. The results confirm that optimized guidelines consistently yield stronger performance when distilled into smaller models.

**History and observation compression.** In Table 14, we report ablations with gpt-4.1 using both history and observation compression. While combining the two compressions achieves larger reductions in peak token usage and dependency, it also leads to substantial performance degradation compared to applying either compression alone.

**Additional guideline optimization step.** We investigate whether running an extra utility maximization step ($\overline{\text{UT}}$) after the standard sequence of utility maximization and compression maximization ($\overline{\text{CO}}$) is beneficial. As shown in Table 14, this additional iteration results in a performance drop, indicating that a single round of optimization is sufficient for effective guideline learning.

**Distilled compressor for observation.** In addition to Section 4.3, we report results for observation compressor distillation in Figure 8. Similar to history compression, the performance is largely preserved after distillation, confirming that ACON enables effective transfer of optimized observation compressors to smaller models.

**Case study: history compression turns failure into success.** A notable case study illustrates how history compression enables a smaller agent to succeed on tasks that would otherwise fail. In the uncompressed trajectory in Example E.2, the gpt-4.1-mini agent repeatedly attempted to use the `file_system` APIs without managing authentication, leading to persistent 401 Unauthorized errors. After compressing the history as in Example E.3, however, the compressed history retained only the essential reasoning steps: the need for both username and password, the importance of passing the returned `access_token` into subsequent calls, and the absence of proxy APIs in the supervisor app.

This compressed context prevented redundant exploration and guided the agent directly to the correct sequence—login with full credentials, capture the token, and provide it explicitly in `show_directory` and `delete_file` calls. As a result, the agent was able to enumerate and

remove all .pdf files in /downloads, a task it had previously failed. This example highlights how compression does not merely shorten history but clarifies critical dependencies, turning a failure trajectory into a successful one.

Table 7: Comparison between ACON and MEM1. The two methods operate under different assumptions regarding model accessibility, training requirements, and architectural coupling.

| Dimension | ACON (ours) | MEM1 (Zhou et al., 2025) |
|---|---|---|
| Is model training not required? | ✓ no model training or weight updates required | ✗ requires RL training on the agent model |
| Can the method work without access to model weights? | ✓ works with open-source and proprietary API models | ✗ requires full model access and gradients |
| Can the agent and compressor be different models? | ✓ supports decoupled design with different model sizes | ✗ reasoning and compression are integrated into a single model |
| Is it possible to use a large agent with a small compressor? | ✓ supports combinations like gpt-4.1 agent + Qwen3-14B compressor | ✗ same model must serve as both agent and compressor |
| Does optimization avoid GPU-based RL cost? | ✓ under $2 for guideline optimization, no GPU needed | ✗ RL policy training requires multiple trajectories and GPU computation |

**Comparison with MEM1.** MEM1 (Zhou et al., 2025) proposes a learnable context compression policy trained jointly with the agent through reinforcement learning. This design couples reasoning and compression within a single trainable model and requires full access to model weights and gradient updates. In contrast, our method can perform optimization entirely at the prompt-level without any weight updates, enabling the agent and compressor to be different models.

This decoupling allows combinations that are not possible in MEM1. For example, one can use a large proprietary model such as gpt-4.1 as the agent while employing a smaller open-source model such as Qwen3-14B as the compressor after distillation, a configuration that MEM1 cannot support due to its unified training requirement. This flexibility makes ACON applicable to both open-source and proprietary API-based models, including settings where model internals are inaccessible. A detailed comparison is summarized in Table 7.

## D QUALITATIVE EXAMPLES

We complement the quantitative results with qualitative illustrations.

**Compression guidelines.** We present examples of compression guidelines before and after optimization in AppWorld. The history compression guideline before optimization is shown in Prompt E.5, the optimized version (UT) in Prompt E.6, and the optimized version (UTCO) in Prompt E.7. Similarly, observation compression guideline examples are provided in Prompt E.8 and Prompt E.9, and the optimized version (UTCO) in Prompt E.10. These comparisons demonstrate that optimization yields more targeted guidelines for compressors.

**Compressed histories.** Compression Example E.1 illustrates history segments before and after guideline optimization in AppWorld with gpt-4.1. The optimized guideline retains a more detailed record of task progress, including variable states and guardrails for the environment. After the compression maximization step (CO), the histories become shorter while still preserving the essential information required for future decision-making. This qualitative evidence demonstrates how our framework improves both the efficiency and effectiveness of context compression, complementing the guideline optimization procedure described in Section 3.3.

We also present Compression Example E.2 for 8-objective QA and Compression Example E.3 for OfficeBench, which confirm that the effects of guideline optimization are consistent across benchmarks.

**Compressed observations.** Compression Example E.4 shows observations before and after guideline optimization in AppWorld. We illustrate the case of printing available APIs for the Spotify app, which produces a lengthy observation. The optimized guideline yields a more structured and faithful representation: whereas naive prompting loses the JSON format and omits the crucial "play_music" API, the optimized version preserves both structure and key functionality necessary to complete the task.

## E  LLM USAGE

We used large language models (LLMs) solely as a writing assistant, for improving grammar and clarity of the paper. No part of the research ideation, experimental design, or analysis relied on LLMs.

---

**Algorithm 1** Alternating Guideline Optimization ($\underline{\text{UT}} \leftrightarrow \overline{\text{CO}}$)

---

**Input:** Training set indices $\mathcal{I}$; fixed agent $\mathcal{M}(\cdot; \theta, \mathcal{P}_{\text{agent}})$; compressor $f(\cdot; \phi, \mathcal{P})$; initial guideline $\mathcal{P}^{(0)}$; tradeoff $\lambda \geq 0$; rounds $R$; candidates $K$

**Output:** Optimized guideline $\mathcal{P}^*$

    **Notation.** For each $i \in \mathcal{I}$ and guideline $\mathcal{P}$:

        baseline (no compression): context sequence $\boldsymbol{H}_i$ with success $r_i^{\text{base}} \in \{0, 1\}$

        compressed: $\boldsymbol{H}_i'(\mathcal{P})$ with success $r_i(\mathcal{P}) \in \{0, 1\}$ and cost $C(\boldsymbol{H}_i'(\mathcal{P})) = \sum_t \mathcal{C}(\boldsymbol{h}_{i,t-1}', o_{i,t}')$

    // 0) Collect baseline contexts (no compression)

1: **for all** $i \in \mathcal{I}$ **do**

2:     Run $\mathcal{M}$ without compression to obtain $\boldsymbol{H}_i$ and $r_i^{\text{base}}$

3: **end for**

4: $\mathcal{I}^+ \leftarrow \{\, i \in \mathcal{I} \mid r_i^{\text{base}} = 1 \,\}$               ▷ indices where baseline succeeds

5: **for** $r = 0$ to $2R - 2$ **step** 2 **do**

    // Stage A: $\underline{\text{UT}}$ (reward-first update using $\boldsymbol{H}$ vs $\boldsymbol{H}'$)

6:     **for all** $i \in \mathcal{I}$ **do**

7:         Run $\mathcal{M}$ with compression $f(\cdot; \phi, \mathcal{P}^{(r)})$ to obtain $\boldsymbol{H}_i'(\mathcal{P}^{(r)}), r_i(\mathcal{P}^{(r)}), C(\boldsymbol{H}_i'(\mathcal{P}^{(r)}))$

8:     **end for**

9:     $\mathcal{D}_{\text{cont}}^{(r)} \leftarrow \{\, (\boldsymbol{H}_i, \boldsymbol{H}_i'(\mathcal{P}^{(r)})) \mid i \in \mathcal{I}^+, \; r_i(\mathcal{P}^{(r)}) = 0 \,\}$

10:     **for all** $(\boldsymbol{H}, \boldsymbol{H}') \in \mathcal{D}_{\text{cont}}^{(r)}$ **do**         ▷ contrastive feedback: what did $\boldsymbol{H}'$ miss vs $\boldsymbol{H}$?

11:         Feedback $\leftarrow$ LLM(FeedbackInstr, $\boldsymbol{H}, \boldsymbol{H}'$)

12:         Append to multiset $\mathcal{F}_{\text{util}}$

13:     **end for**

14:     $\{\mathcal{P}_k^{(r+1)}\}_{k=1}^K \leftarrow$ LLM(UpdateInstr, $\mathcal{P}^{(r)}, \|_{f \in \mathcal{F}_{\text{util}}} f$)         // $\|$: concatenation

15:     **Select by reward:** evaluate on a held-out subset of $\mathcal{I}^+$ and pick

$$k_{\text{util}}^* \leftarrow \arg\max_k \; \text{SuccessRate}\big(\{r_i(\mathcal{P}_k^{(r+1)})\}_{i \in \mathcal{I}^+}\big)$$

16:     $\mathcal{P}_{\text{util}}^{(r+1)} \leftarrow \mathcal{P}_{k_{\text{util}}^*}^{(r+1)}$

    // Stage B: $\overline{\text{CO}}$ (cost-minimizing refinement using only $\boldsymbol{H}'$)

17:     **for all** $i \in \mathcal{I}$ **do**

18:         Using $\mathcal{P}_{\text{util}}^{(r+1)}$, obtain $\boldsymbol{H}_i', r_i, C(\boldsymbol{H}_i')$

19:     **end for**

20:     $\mathcal{D}_{\text{succ}}^{(r)} \leftarrow \{\, \boldsymbol{H}_i' \mid r_i = 1 \,\}$

21:     **for all** $\boldsymbol{H}' \in \mathcal{D}_{\text{succ}}^{(r)}$ **do**         ▷ find redundant spans within $\boldsymbol{H}'$

22:         CompFeedback $\leftarrow$ LLM(CompressInstr, $\boldsymbol{H}'$)

23:         Append to multiset $\mathcal{F}_{\text{comp}}$

24:     **end for**

25:     $\{\tilde{\mathcal{P}}_k^{(r+2)}\}_{k=1}^K \leftarrow$ LLM(UpdateInstr_Compress, $\mathcal{P}_{\text{util}}^{(r+1)}, \|_{f \in \mathcal{F}_{\text{comp}}} f$)

26:     **Select by reward–cost:** evaluate on a held-out split of $\mathcal{I}$ and pick

$$k_{\text{comp}}^* \leftarrow \arg\max_k \; \Big( \text{SuccessRate}(\{r_i(\tilde{\mathcal{P}}_k^{(r+2)})\}) \; - \; \lambda \cdot \text{NormCost}(\{C(\boldsymbol{H}_i'(\tilde{\mathcal{P}}_k^{(r+2)}))\}) \Big)$$

27:     $\mathcal{P}^{(r+2)} \leftarrow \tilde{\mathcal{P}}_{k_{\text{comp}}^*}^{(r+2)}$

28:     **if** early-stop criterion satisfied **then**         ▷ e.g., success/cost convergence or budget met

29:         **break**

30:     **end if**

31: **end for**

32: $\mathcal{P}^* \leftarrow \mathcal{P}^{(r+2)}$

33:

34: **return** $\mathcal{P}^*$

---

Table 8: Detailed results on **OfficeBench** benchmark. We report accuracy (%), and efficiency metrics: average steps, peak input tokens ($10^3$), and dependency ($10^6$) for Average and each difficulty level. Best values are in bold. Rows in blue background indicate the results from ours .

| Method | Average (All) | | | | Level 1 (1-app, 42) | | | Level 2 (2-app, 22) | | | Level 3 (3-app, 31) | | |
|---|---|---|---|---|---|---|---|---|---|---|---|---|---|
| | Acc. ↑ | Steps ↓ | Peak ↓ | Dep. ↓ | Acc. ↑ | Peak ↓ | Dep. ↓ | Acc. ↑ | Peak ↓ | Dep. ↓ | Acc. ↑ | Peak ↓ | Dep. ↓ |
| **Agent: gpt-4.1 / Compressor: gpt-4.1** | | | | | | | | | | | | | |
| No Compression | **76.84** | 11.52 | 7.27 | 4.43 | **92.86** | 6.23 | 4.05 | 77.27 | 6.14 | 1.81 | 54.84 | 8.37 | 6.08 |
| **History Compression** | | | | | | | | | | | | | |
| FIFO | 67.37 | 12.26 | **4.02** | 2.64 | 83.33 | 4.19 | 0.72 | 63.64 | **3.51** | **1.01** | 48.39 | **4.23** | 4.39 |
| Retrieval | 65.26 | 16.20 | 4.33 | 2.06 | 85.71 | 4.35 | 0.84 | 63.64 | 3.52 | 1.37 | 38.71 | 4.78 | 2.99 |
| LLMLingua | 70.53 | 10.89 | 4.65 | 1.85 | 83.33 | 4.17 | **0.67** | 68.18 | 4.61 | 1.18 | 54.84 | 4.88 | 2.74 |
| Prompting | 71.58 | **10.13** | 4.40 | **1.10** | 85.71 | 4.18 | 0.81 | 77.27 | 4.53 | 1.08 | 48.39 | 4.42 | **1.23** |
| ACON UT | 74.74 | 13.13 | 4.93 | 3.85 | 85.71 | 4.71 | 6.89 | 72.73 | 4.64 | 1.44 | 61.29 | 5.19 | 3.89 |
| ACON UTCO | 72.63 | 11.54 | 4.54 | 1.91 | 88.10 | 3.92 | 0.76 | 72.73 | 4.72 | 1.16 | 51.61 | 4.71 | 2.84 |
| **Observation Compression** | | | | | | | | | | | | | |
| LLMLingua | 71.58 | 11.89 | 7.38 | 6.14 | 80.95 | 7.35 | 12.40 | 72.73 | 6.31 | 2.11 | **58.06** | 7.99 | 5.70 |
| Prompting | 55.79 | 12.24 | 6.44 | 2.68 | 78.57 | 4.51 | 0.98 | 50.00 | 6.98 | 2.61 | 29.03 | 6.98 | 3.46 |
| ACON UT | 73.68 | 10.83 | 6.55 | 3.85 | 90.48 | 6.57 | 8.02 | 77.27 | 6.11 | 1.97 | 48.39 | 6.80 | 3.10 |
| ACON UTCO | 72.63 | 10.28 | 6.17 | 2.88 | 88.10 | 4.75 | 0.82 | 72.73 | 6.41 | 2.09 | 51.61 | 6.65 | 4.22 |

Table 9: Results across different difficulty levels on **Appworld** benchmark (test-normal) with gpt-4.1-mini. We adopt the same compression guidelines as those used in the gpt-4.1 experiments. Each block reports accuracy (task goal completion score), average steps, average peak input tokens ($10^3$), and average dependency ($10^6$) for agents. Best results in each column are highlighted in bold. Rows in blue background indicate the results from ours .

| Method | Average (168) | | | | Easy (57) | | | Medium (48) | | | Hard (63) | | |
|---|---|---|---|---|---|---|---|---|---|---|---|---|---|
| | Acc. ↑ | Steps ↓ | Peak ↓ | Dep.↓ | Acc. ↑ | Peak ↓ | Dep.↓ | Acc. ↑ | Peak ↓ | Dep.↓ | Acc. ↑ | Peak ↓ | Dep.↓ |
| **Agent: gpt-4.1-mini / Compressor: gpt-4.1-mini** | | | | | | | | | | | | | |
| No compression | 35.7 | 18.14 | 8.55 | 5.07 | 56.1 | 6.45 | 3.72 | 31.2 | 8.31 | 4.79 | 20.6 | 10.64 | 9.18 |
| **History Compression** | | | | | | | | | | | | | |
| FIFO | 39.3 | 30.39 | 6.18 | 5.24 | 75.4 | 4.76 | 2.66 | 35.4 | 5.33 | 4.81 | 9.5 | 8.10 | 7.91 |
| Retrieval | 14.9 | 40.18 | 7.49 | 5.95 | 36.8 | 7.10 | 4.29 | 8.3 | 7.44 | 6.80 | 0.0 | 7.89 | 6.81 |
| LLMLingua | 36.3 | 28.41 | 7.24 | 6.65 | 66.7 | 6.96 | 3.84 | 33.3 | 7.05 | 7.60 | 11.1 | 7.62 | 8.47 |
| Prompting | 35.7 | 24.98 | 6.56 | 4.95 | 64.9 | 5.96 | 2.90 | 27.1 | 6.65 | 5.35 | 15.9 | 6.84 | 6.49 |
| ACON UT | 42.3 | 22.46 | 6.51 | 5.48 | 64.9 | 5.87 | 2.62 | 37.5 | 7.18 | 5.22 | 25.4 | 7.18 | 8.25 |
| ACON UTCO | 32.7 | 24.27 | 6.99 | 4.97 | 57.9 | 7.50 | 2.77 | 33.3 | 8.45 | 4.99 | 9.5 | 6.95 | 6.97 |
| **Observation Compression** | | | | | | | | | | | | | |
| LLMLingua | 25.6 | 20.75 | 8.04 | 8.21 | 38.6 | 6.13 | 3.03 | 27.1 | 8.74 | 13.78 | 12.7 | 9.24 | 8.65 |
| Prompting | 33.9 | 16.71 | 6.04 | 3.87 | 59.7 | 5.21 | 3.41 | 33.3 | 5.99 | 3.27 | 11.1 | 6.83 | 4.74 |
| ACON UT | 33.9 | 16.78 | 6.86 | 4.58 | 59.7 | 5.44 | 2.93 | 33.3 | 7.13 | 4.26 | 11.1 | 7.97 | 6.38 |
| ACON UTCO | 27.4 | 17.89 | 6.37 | 4.44 | 40.4 | 5.18 | 2.40 | 35.4 | 6.84 | 5.03 | 9.5 | 7.09 | 5.82 |

Table 10: Detailed results on **OfficeBench** benchmark with gpt-4.1-mini. We adopt the same compression guidelines as those used in the gpt-4.1 experiments. We report accuracy (%), and efficiency metrics: average steps, peak input tokens ($10^3$), and dependency ($10^6$) for Average and each difficulty level. Rows in blue background indicate the results from ours .

| Method | Average (All) | | | | Level 1 (1-app, 42) | | | Level 2 (2-app, 22) | | | Level 3 (3-app, 31) | | |
|---|---|---|---|---|---|---|---|---|---|---|---|---|---|
| | Acc. ↑ | Steps ↓ | Peak ↓ | Dep. ↓ | Acc. ↑ | Peak ↓ | Dep. ↓ | Acc. ↑ | Peak ↓ | Dep. ↓ | Acc. ↑ | Peak ↓ | Dep. ↓ |
| **Agent: gpt-4.1-mini / Compressor: gpt-4.1-mini** | | | | | | | | | | | | | |
| No Compression | 72.63 | 11.96 | 7.36 | 3.92 | 88.10 | 6.66 | 4.29 | 68.18 | 4.97 | 1.01 | 54.84 | 9.02 | 5.40 |
| **History Compression** | | | | | | | | | | | | | |
| FIFO | 65.26 | 10.91 | 4.03 | 1.46 | 83.33 | 4.10 | 0.78 | 59.09 | 3.69 | 0.96 | 45.16 | 4.19 | 2.03 |
| Retrieval | 67.37 | 14.46 | 4.55 | 2.74 | 85.71 | 5.85 | 5.86 | 59.09 | 3.47 | 0.87 | 48.39 | 4.59 | 2.45 |
| LLMLingua | 67.39 | 11.59 | 4.90 | 2.18 | 87.18 | 4.31 | 3.87 | 59.09 | 4.58 | 0.92 | 48.39 | 5.34 | 2.17 |
| Prompting | 71.58 | 11.78 | 4.93 | 3.10 | 85.71 | 4.73 | 4.75 | 72.73 | 4.40 | 0.86 | 51.61 | 5.32 | 3.06 |
| ACON | 73.68 | 12.41 | 4.82 | 1.96 | 88.10 | 4.12 | 0.83 | 68.18 | 4.39 | 0.86 | 58.06 | 5.37 | 3.07 |
| **Observation Compression** | | | | | | | | | | | | | |
| LLMLingua | 66.32 | 11.02 | 6.34 | 2.40 | 78.57 | 6.09 | 2.12 | 63.64 | 4.82 | 0.97 | 51.61 | 7.30 | 3.34 |
| Prompting | 73.68 | 11.43 | 6.45 | 2.62 | 88.10 | 4.82 | 1.44 | 72.73 | 4.95 | 1.06 | 54.84 | 8.01 | 4.01 |
| ACON | 71.58 | 10.96 | 6.00 | 2.19 | 88.10 | 4.45 | 1.06 | 63.64 | 4.89 | 1.00 | 54.84 | 7.30 | 3.36 |

Table 11: Results on **8-objective QA** benchmark with gpt-4.1-mini. We adopt the same compression guidelines as those used in the gpt-4.1 experiments. We report EM/F1 and efficiency metrics (Steps, Peak input tokens ($10^3$), and Dependency ($10^6$)).

| Method | EM ↑ | F1 ↑ | Steps ↓ | Peak ↓ | Dep. ↓ |
|---|---|---|---|---|---|
| **Agent:** gpt-4.1-mini / **Compressor:** gpt-4.1-mini | | | | | |
| No compression | 0.330 | 0.436 | 19.80 | 12.93 | 5.63 |
| **History Compression** | | | | | |
| FIFO | 0.024 | 0.031 | 28.45 | 5.33 | 3.89 |
| Retrieval | 0.143 | 0.190 | 26.90 | 5.34 | 3.55 |
| LLMLingua | 0.140 | 0.194 | 25.24 | 6.69 | 3.92 |
| Prompting | 0.149 | 0.207 | 25.27 | 4.85 | 2.44 |
| ACON | 0.238 | 0.325 | 21.05 | 4.78 | 2.03 |
| ACON (iter2) | 0.248 | 0.353 | 19.18 | 4.79 | 1.79 |
| **Observation Compression** | | | | | |
| LLMLingua | 0.316 | 0.430 | 15.96 | 5.54 | 1.60 |
| Prompting | 0.282 | 0.402 | 11.71 | 3.91 | 0.65 |
| ACON | 0.323 | 0.434 | 14.42 | 4.71 | 1.10 |
| ACON (iter2) | 0.316 | 0.443 | 11.69 | 3.97 | 0.63 |

Table 12: Results across different difficulty levels on **AppWorld** benchmark (test-normal) with gpt-5-chat. We adopt the same compression guidelines as those used in the gpt-4.1 experiments. Each block reports accuracy (task goal completion score), steps, peak input tokens ($10^3$), and dependency ($10^6$) for agents. Best results in each column are highlighted in bold. Rows in blue background indicate the results from **ours** .

| Method | Average (168) | | | | Easy (57) | | | Medium (48) | | | Hard (63) | | |
|---|---|---|---|---|---|---|---|---|---|---|---|---|---|
| | Acc. ↑ | Steps ↓ | Peak ↓ | Dep.↓ | Acc. ↑ | Peak ↓ | Dep.↓ | Acc. ↑ | Peak ↓ | Dep.↓ | Acc. ↑ | Peak ↓ | Dep.↓ |
| **Agent:** gpt-5-chat / **Compressor:** gpt-5-chat | | | | | | | | | | | | | |
| No compression | 66.7 | 16.45 | 9.67 | 4.78 | 89.5 | 7.55 | 2.31 | 64.6 | 9.58 | 4.13 | 47.6 | 11.67 | 7.51 |
| **History Compression** | | | | | | | | | | | | | |
| FIFO (last-5) | 46.4 | 30.61 | 6.81 | 4.85 | 79.0 | 5.21 | 2.10 | 43.8 | 6.82 | 5.50 | 19.1 | 8.24 | 6.84 |
| Prompting | 58.9 | 22.24 | 7.46 | 4.02 | 82.5 | 7.15 | 2.13 | 66.7 | 7.19 | 3.69 | 31.8 | 7.93 | 5.97 |
| ACON UT | 58.3 | 20.15 | 6.97 | 3.74 | 80.7 | 6.66 | 2.04 | 66.7 | 7.08 | 3.40 | 31.8 | 7.16 | 5.54 |
| ACON UTCO | 62.5 | 22.29 | 7.26 | 3.85 | 86.0 | 6.44 | 2.04 | 72.9 | 6.98 | 3.93 | 33.3 | 8.20 | 5.42 |
| **Observation Compression** | | | | | | | | | | | | | |
| Prompting | 60.1 | 17.39 | 6.50 | 3.72 | 80.7 | 4.98 | 1.72 | 68.8 | 6.40 | 3.48 | 34.9 | 7.96 | 5.70 |
| ACON UT | 65.5 | 17.16 | 7.58 | 3.96 | 84.2 | 5.62 | 1.94 | 68.8 | 7.49 | 3.46 | 46.0 | 9.41 | 6.16 |
| ACON UTCO | 62.5 | 18.21 | 7.21 | 4.24 | 80.7 | 5.52 | 2.02 | 70.8 | 7.18 | 3.69 | 39.7 | 8.76 | 6.67 |
| **History + Observation Compression** | | | | | | | | | | | | | |
| ACON UT | 63.1 | 20.02 | 5.89 | 3.63 | 77.2 | 5.27 | 1.92 | 77.1 | 6.03 | 3.52 | 39.7 | 6.35 | 5.28 |
| ACON UTCO | 58.9 | 22.90 | 5.83 | 4.07 | 80.7 | 5.35 | 1.94 | 77.1 | 5.94 | 3.56 | 25.4 | 6.17 | 6.39 |

Table 13: Results across different difficulty levels on **AppWorld** with **distilled compressors**. We report accuracy (task goal completion score), average steps, peak input tokens ($10^3$), and dependency ($10^6$). For all compressors, we use the optimized compression guideline after the utilization maximization UT step. 'Fine-tune' means that we fine-tune small models with outputs from naive prompt before compression guideline optimization.

| Method | Average | | | | Easy | | | Medium | | | Hard | | |
|---|---|---|---|---|---|---|---|---|---|---|---|---|---|
| | Acc. ↑ | Steps ↓ | Peak ↓ | Dep.↓ | Acc. ↑ | Peak ↓ | Dep.↓ | Acc. ↑ | Peak ↓ | Dep.↓ | Acc. ↑ | Peak ↓ | Dep.↓ |
| **Agent:** gpt-4.1 / **Compressor:** gpt-4.1-mini or Distilled models (Qwen3, Phi-4) | | | | | | | | | | | | | |
| **History Compression** | | | | | | | | | | | | | |
| Prompting (gpt-4.1-mini) | 39.3 | 23.61 | 7.03 | 5.19 | 64.9 | 6.64 | 3.17 | 35.4 | 7.63 | 5.42 | 19.1 | 6.93 | 6.84 |
| ACON (gpt-4.1-mini) | 47.6 | 21.46 | 7.25 | 5.24 | 75.4 | 6.75 | 2.84 | 35.4 | 7.25 | 5.36 | 31.8 | 7.70 | 7.32 |
| Fine-tune (Qwen3-14B) | 44.6 | 24.16 | 7.16 | 4.95 | 71.9 | 6.79 | 2.88 | 43.8 | 7.39 | 4.88 | 20.6 | 7.33 | 6.88 |
| ACON (Qwen3-14B) | 50.0 | 21.72 | 6.83 | 4.80 | 79.0 | 6.42 | 2.54 | 50.0 | 6.87 | 4.89 | 23.8 | 7.17 | 6.79 |
| ACON (Qwen3-8B) | 47.0 | 21.58 | 6.98 | 4.76 | 71.9 | 6.64 | 2.93 | 37.5 | 7.24 | 4.67 | 31.8 | 7.09 | 6.48 |
| ACON (Phi-4) | 44.6 | 21.19 | 7.24 | 4.76 | 68.4 | 7.33 | 2.75 | 39.6 | 7.12 | 4.16 | 27.0 | 7.26 | 7.04 |
| **Observation Compression** | | | | | | | | | | | | | |
| Prompting (gpt-4.1-mini) | 44.0 | 16.67 | 6.84 | 4.30 | 71.9 | 5.08 | 2.19 | 35.4 | 6.72 | 3.77 | 25.4 | 8.53 | 6.61 |
| ACON (gpt-4.1-mini) | 48.2 | 18.00 | 8.66 | 6.62 | 71.9 | 6.05 | 2.60 | 37.5 | 9.23 | 7.41 | 34.9 | 10.60 | 9.65 |
| Fine-tune (Qwen3-14B) | 40.5 | 17.71 | 6.64 | 4.38 | 64.9 | 4.91 | 1.97 | 31.2 | 6.72 | 4.05 | 25.4 | 8.16 | 6.81 |
| ACON (Qwen3-14B) | 56.5 | 16.78 | 7.57 | 5.06 | 82.5 | 5.69 | 2.20 | 54.2 | 7.39 | 4.46 | 34.9 | 9.40 | 8.10 |
| ACON (Qwen3-8B) | 48.2 | 16.10 | 7.33 | 4.82 | 71.9 | 5.49 | 2.03 | 50.0 | 7.20 | 4.20 | 25.4 | 9.10 | 7.82 |
| ACON (Phi-4) | 50.6 | 16.88 | 7.88 | 5.41 | 77.2 | 5.85 | 2.88 | 52.1 | 7.75 | 4.77 | 25.4 | 9.83 | 8.18 |

Table 14: Additional results for additional guideline optimization step and unified compression on **Appworld** benchmark (test-normal). Each block reports accuracy (task goal completion score), steps, peak input tokens ($10^3$), and dependency ($10^6$) for agents. Best results in each column are highlighted in bold. Rows in blue background indicate the results from **ours** .

| Method | Average (168) | | | | Easy (57) | | | Medium (48) | | | Hard (63) | | |
|---|---|---|---|---|---|---|---|---|---|---|---|---|---|
| | Acc. ↑ | Steps ↓ | Peak ↓ | Dep.↓ | Acc. ↑ | Peak ↓ | Dep.↓ | Acc. ↑ | Peak ↓ | Dep.↓ | Acc. ↑ | Peak ↓ | Dep.↓ |
| **Agent:** gpt-4.1 / **Compressor:** gpt-4.1 | | | | | | | | | | | | | |
| **History Compression** | | | | | | | | | | | | | |
| ACON UTCOUT | 47.0 | 22.28 | 7.22 | 4.66 | 68.4 | 7.01 | 2.69 | 58.3 | 7.16 | 4.39 | 19.1 | 7.45 | 6.65 |
| **History + Observation Compression** | | | | | | | | | | | | | |
| Prompting | 36.3 | 19.33 | 5.38 | 3.44 | 71.9 | 4.87 | 1.80 | 21.6 | 5.63 | 3.60 | 14.3 | **5.64** | 4.79 |
| ACON | 45.8 | 20.32 | 5.85 | 4.26 | 75.4 | 5.29 | 2.07 | 39.6 | 6.15 | 4.29 | 23.8 | 6.12 | 6.21 |
| ACON UTCO | 44.6 | 21.75 | 5.90 | 4.98 | 77.2 | 5.50 | 2.33 | 39.6 | 6.18 | 3.80 | 19.1 | 6.18 | 8.28 |

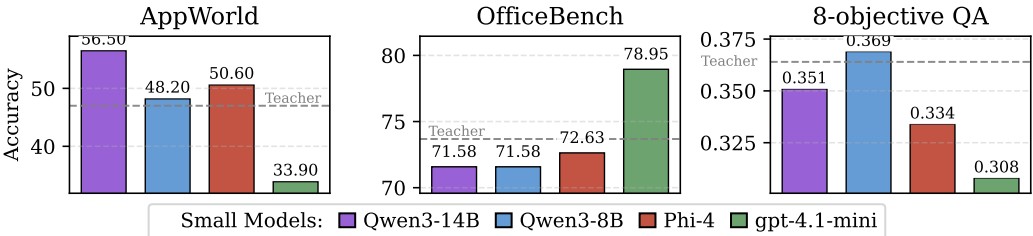

Figure 8: **Results of distilled compressors on observation compression** with gpt-4.1 as the agent. Student models (Qwen3-14B, Qwen3-8B, Phi-4) are distilled from gpt-4.1 compressor using the optimized compression guideline after UT step, and evaluated across all benchmarks. We also include result with gpt-4.1-mini without distillation for comparison.

---

**Prompt E.1: Prompt for analysis before prompt optimization (utility step)**

```
You are an expert agent trajectory auditor.

Analyze why the HISTORY-OPTIMIZED agent failed OR became
    significantly less efficient while the BASELINE succeeded.
You are given:
- task_name: {{ task_name }}
- Baseline full history (single continuous session)
- Optimized history split into multiple sessions where each new
    session starts with a fresh system + user prompt and an
    injected <HISTORY_SUMMARY> summarizing earlier interactions.
- baseline_success={{ baseline_success }} optimized_success={{
    optimized_success }}
- baseline_env_steps={{ baseline_env_steps | default('null') }}
    optimized_env_steps={{ optimized_env_steps | default('null')
    }} step_ratio={{ step_ratio | default('null') }}
    performance_regression={{ performance_regression | default('
    false') }}

Goals:
1. Determine whether summarization / session resetting removed,
    distorted, delayed, or bloated reasoning causing failure OR
    inflated step count (> threshold factor of baseline).
2. Identify the FIRST divergence point where the optimized
    trajectory meaningfully deviates from the successful &
    efficient baseline path.
3. Categorize root causes (e.g., Missing Critical Fact,
    Incorrect Summary, Lost Variable/State, Unnecessary Re-
    discovery, Instruction Drift, API Misuse, Premature
    Completion, Token Truncation, Inefficient Looping, Redundant
     API Calls, Over-Exploration, Other).
4. Extract concrete evidence snippets (quote exact lines) from
    baseline vs optimized showing:
    - Critical facts present in baseline but absent/altered in
    optimized (esp. after a session boundary)
    - Summary inaccuracies (baseline ground truth vs summary text
    )
    - Redundant or looping action patterns causing step inflation
    .
5. Suggest precise remediation strategies: summary style changes
    , retain variable/value tables, move session boundaries,
    guardrail prompts, caching, early-exit heuristics, loop
    detection, etc.
6. Provide a reliability_score (0.0-1.0) reflecting confidence
    in your causal attribution.
7. If performance_regression==true, analyze efficiency
    degradation even if optimized_success==true.

Output STRICTLY valid JSON object with keys:
{
  "task_name": str,
  "divergence_step_description": str,
  "root_cause_categories": [str, ...],
  "missing_or_distorted_facts": [ {"baseline": str, "
    optimized_context_absent_or_changed": str, "impact": str} ],
  "summary_inaccuracies": [ {"summary_excerpt": str, "issue_type
    ": str, "correct_baseline_reference": str, "impact": str} ],
  "lost_state_variables": [ {"name_or_pattern": str, "
    baseline_evidence": str, "optimized_issue": str} ],
  "api_or_action_errors": [ {"optimized_step_excerpt": str, "
    error_type": str, "improvement": str} ],
```

```
    "inefficiency_patterns": [ {"pattern": str, "evidence_excerpt
      ": str, "excess_steps": int, "cause": str, "remediation":
      str} ],
    "timeline_of_divergence": [ {"phase": str, "optimized_excerpt
      ": str, "baseline_contrast": str, "effect": str} ],
    "performance_regression": bool,
    "baseline_env_steps": int | null,
    "optimized_env_steps": int | null,
    "step_ratio": float | null,
    "remediation_recommendations": [ str, ... ],
    "recovery_opportunities_missed": [ {"optimized_excerpt": str,
      "missed_fix_action": str} ],
    "reliability_score": float,
    "concise_failure_mechanism_summary": str
}

If some sections have no data, use an empty list. For non-
    applicable numeric fields use null.
Do NOT include any extra commentary outside JSON.

---
BASELINE_HISTORY_START
{{ baseline_history }}
BASELINE_HISTORY_END

OPTIMIZED_MULTI_SESSION_HISTORY_START
{{ optimized_history }}
OPTIMIZED_MULTI_SESSION_HISTORY_END

Failure or performance report / metadata (may be null):
{{ failure_report }}

Proceed with rigorous comparison.
```

**Prompt E.2: Prompt for prompt optimization after analysis (utility step)**

```
You are an expert prompt engineer tasked with refining a HISTORY
    SUMMARIZATION prompt.
Rewrite the ORIGINAL PROMPT to reduce length of the HISTORY
    SUMMARY while preserving factual continuity for the next
    session.
Ground all changes in the PER-SAMPLE REDUCTION SIGNALS below. Do
    not aggregate across samples; use the patterns and rules as
    -is.

Constraints:
- Keep all Jinja placeholders, variable names, and structure
    intact where possible.
- Add explicit, concrete rules that prevent verbosity and retain
    essential state.
- Do not include literal values from prior content; refer to
    variable names only.
- Output ONLY the improved prompt template (no extra commentary)
    .

Context (samples below are the only ground truth signals to use)
    :
- Average original summary size (chars) across sampled set: {{
    avg_orig_chars }}

{% for s in samples %}
```

```
===== SAMPLE {{ loop.index0 }} =====
- Task/Session: {{ s.task_label }} / {{ s.session or 'unknown-
    session' }}
- Analysis Overview:
{% if s.overview %}
{% for k, v in s.overview.items() %}  - {{ k }}: {{ v }}
{% endfor %}
{% else %}  - (none provided)
{% endif %}

- Removals (patterns -> action):
{% for r in s.removals %}  - [{{ r.category | default('unknown')
    }}] {{ r.pattern | default('') }} -> {{ r.action | default
    ('drop') }}
{% endfor %}

- KEEP examples (evidence-driven essentials):
{% for k in s.keeps %}  - Reason: {{ k.reason | default('') }} |
    Evidence: {{ k.evidence_spans | default([]) | join('; ') }}
{% endfor %}

- Summary Rules:
{% for rule in s.rules %}  - {{ rule }}
{% endfor %}

{% endfor %}

Original Prompt Template (verbatim between markers):
<<<ORIGINAL_PROMPT>>>
{{ original_prompt }}
<<<ORIGINAL_PROMPT>>>

Output only the improved prompt template text, ready to be used
    as a Jinja template.
```

**Prompt E.3: Prompt for analysis before prompt optimization (compression step)**

```
You are an expert prompt engineer tasked with refining a HISTORY
    SUMMARIZATION prompt.
Rewrite the ORIGINAL PROMPT to reduce length of the HISTORY
    SUMMARY while preserving factual continuity for the next
    session.
Ground all changes in the PER-SAMPLE REDUCTION SIGNALS below. Do
    not aggregate across samples; use the patterns and rules as
    -is.

Constraints:
- Keep all Jinja placeholders, variable names, and structure
    intact where possible.
- Add explicit, concrete rules that prevent verbosity and retain
    essential state.
- Do not include literal values from prior content; refer to
    variable names only.
- Output ONLY the improved prompt template (no extra commentary)
    .

Context (samples below are the only ground truth signals to use)
    :
- Average original summary size (chars) across sampled set: {{
    avg_orig_chars }}
```

```
{% for s in samples %}
===== SAMPLE {{ loop.index0 }} =====
- Task/Session: {{ s.task_label }} / {{ s.session or 'unknown-
    session' }}
- Analysis Overview:
{% if s.overview %}
{% for k, v in s.overview.items() %}  - {{ k }}: {{ v }}
{% endfor %}
{% else %}  - (none provided)
{% endif %}

- Removals (patterns -> action):
{% for r in s.removals %}  - [{{ r.category | default('unknown')
    }}] {{ r.pattern | default('') }} -> {{ r.action | default
    ('drop') }}
{% endfor %}

- KEEP examples (evidence-driven essentials):
{% for k in s.keeps %}  - Reason: {{ k.reason | default('') }} |
    Evidence: {{ k.evidence_spans | default([]) | join('; ') }}
{% endfor %}

- Summary Rules:
{% for rule in s.rules %}  - {{ rule }}
{% endfor %}

{% endfor %}

Original Prompt Template (verbatim between markers):
<<<ORIGINAL_PROMPT>>>
{{ original_prompt }}
<<<ORIGINAL_PROMPT>>>

Output only the improved prompt template text, ready to be used
    as a Jinja template.
```

**Prompt E.4: Prompt for analysis before prompt optimization (compression step)**

```
You are an expert prompt engineer tasked with refining a HISTORY
    SUMMARIZATION prompt.
Rewrite the ORIGINAL PROMPT to reduce length of the HISTORY
    SUMMARY while preserving factual continuity for the next
    session.
Ground all changes in the PER-SAMPLE REDUCTION SIGNALS below. Do
    not aggregate across samples; use the patterns and rules as
    -is.

Constraints:
- Keep all Jinja placeholders, variable names, and structure
    intact where possible.
- Add explicit, concrete rules that prevent verbosity and retain
    essential state.
- Do not include literal values from prior content; refer to
    variable names only.
- Output ONLY the improved prompt template (no extra commentary)
    .

Context (samples below are the only ground truth signals to use)
    :
- Average original summary size (chars) across sampled set: {{
    avg_orig_chars }}
```

```
{% for s in samples %}
===== SAMPLE {{ loop.index0 }} =====
- Task/Session: {{ s.task_label }} / {{ s.session or 'unknown-
    session' }}
- Analysis Overview:
{% if s.overview %}
{% for k, v in s.overview.items() %}  - {{ k }}: {{ v }}
{% endfor %}
{% else %}  - (none provided)
{% endif %}

- Removals (patterns -> action):
{% for r in s.removals %}  - [{{ r.category | default('unknown')
    }}] {{ r.pattern | default('') }} -> {{ r.action | default
    ('drop') }}
{% endfor %}

- KEEP examples (evidence-driven essentials):
{% for k in s.keeps %}  - Reason: {{ k.reason | default('') }} |
    Evidence: {{ k.evidence_spans | default([]) | join('; ') }}
{% endfor %}

- Summary Rules:
{% for rule in s.rules %}  - {{ rule }}
{% endfor %}

{% endfor %}

Original Prompt Template (verbatim between markers):
<<<ORIGINAL_PROMPT>>>
{{ original_prompt }}
<<<ORIGINAL_PROMPT>>>

Output only the improved prompt template text, ready to be used
    as a Jinja template.
```

**Prompt E.5: AppWorld Prompt for history compression before optimization**

```
You are maintaining a structured context-aware summary for a
    productivity agent. You will be given the user instruction
    for the agent, a list of interactions corresponding to
    actions taken by the agent, and the most recent previous
    summary if one exists. Produce the following:

### REASONING
Summarize key progress, decisions made, important observed
    outcomes, and rationale behind actions taken so far. Include
     how earlier steps influenced later ones and why certain
    data is retained in the summary.

### COMPLETED
List completed subtasks or successful outcomes, with brief
    results if applicable.

---

## [Information Source]

### USER INSTRUCTION
```

```
{{ task }}

## [PREVIOUS SUMMARY] (if any)

{{ prev_summary }}

## [HISTORY OF INTERACTIONS]

{{ history }}

---

## PRIORITIZE

1. Keep all sections relevant and concise.
2. Use reusable structured formats when summarizing artifacts.
3. Ensure agent can resume task with no loss of information.
4. Include key info from errors or failed attempts to prevent
    repeated mistakes.
5. Preserve all essential artifacts and data needed to complete
    the task.

---

### [Output Format]

Do **not** include the input or any additional explanation. Only
    return the formatted summary.
```

**Prompt E.6: AppWorld Prompt for history compression after optimization (UT)**

```
You maintain a compact, state-preserving HISTORY_SUMMARY for a
    multi-session agent.

Input:
[USER INSTRUCTION] {{ task }}
[PREVIOUS SUMMARY] {{ prev_summary }}
[HISTORY OF INTERACTIONS] {{ history }}

Create the following sections-use the exact headings and order:

<HISTORY_SUMMARY>

1. REASONING
   - Key progress, decisions, outcomes, and their rationale.
   - Note how earlier steps influence later ones.

2. VARS
   | name | value | purpose |
   |------|-------|---------|
   Record every runtime value the next session must re-declare (
    tokens, ids, lists, last page_index/page_limit, etc.).

3. TODO
   List pending actions with enough detail to execute directly.

4. COMPLETED
   Bullet list of finished subtasks with brief results.

5. GUARDRAILS
   Short reminders that prevent repeat errors, e.g.
```

```
    - Memory resets; re-create VARS before use.
    - Paginate until empty page.
    - Validate API parameters against spec.
    - Avoid redundant logins or doc look-ups.

Requirements:
- Be concise-bullets and tables preferred; no extraneous prose.
- Preserve all essential facts, parameters, and artifacts; omit
    nothing critical.
- Include errors only if they inform future avoidance.
- Do not output the input or any commentary-return only <
    HISTORY_SUMMARY>.
```

**Prompt E.7: AppWorld Prompt for history compression after optimization (UTCO)**

```
You maintain a compact, state-preserving HISTORY_SUMMARY for a
    multi-session agent.

Input:
[USER INSTRUCTION] {{ task }}
[PREVIOUS SUMMARY] {{ prev_summary }}
[HISTORY OF INTERACTIONS] {{ history }}

Summary Compression Rules:
- Collapse multi-bullet narratives into <=2 concise sentences.
- Replace repetitive step logs with one summarizing phrase.
- Truncate long token/credential strings to "<token>" unless
    verbatim reuse is required.
- Remove unused/expired credentials, page_index/page_limit,
    verbose API dumps, and table borders.
- Shrink GUARDRAILS to one bullet unless multiple items are
    still critical.
- Delete tool/API log output, greetings, meta prose, and section
     headers that no longer contain content.
- Keep only variables actively referenced in upcoming steps;
    list each once in VARS.
- Reference removal categories [repetition], [tool-logs], [meta
    ], [formatting] to prune similar lines.
- Preserve factual continuity; never invent or alter state
    variables.
- Target summaries well under {{ max_chars | default(1500) }}
    characters.

Critical Essentials:
Always keep evidence-driven items required next session (e.g.,
    tokens, ids, emails, amounts, lists, paths, description
    strings, brief task status).

Output EXACTLY the following structure---nothing more:

<HISTORY_SUMMARY>

1. REASONING
   One brief paragraph on key progress and rationale.

2. VARS
   key=value pairs, comma-separated; only still-needed runtime
    values.

3. TODO
```

```
      Bulleted next actions (<=5).

   4. COMPLETED
      Bulleted finished subtasks (<=5).

   5. GUARDRAILS
      Single concise bullet, or omit if none.

   Return only the <HISTORY_SUMMARY> block---no additional
       commentary or input echoes.
```

**Prompt E.8: AppWorld Prompt for observation compression before optimization**

```
Your task is to generate a "Reasoning" and a "Refined
    Observation" based on the inputs below.

In the "Reasoning", analyze the user instruction and history to
    identify what information from the current observation is
    necessary to complete the remaining steps.
Think about what parts can be summarized or transformed to
    reduce length, while ensuring that future actions can still
    be executed based on the refined observation alone.

In the "Refined Observation", include only the information that
    is minimal but sufficient for the next steps.

[Information source]
# User Instruction
{{ task }}

# History of interactions
{{ history }}

# Observation at the current time step
{{ observation }}

[Output format]
# Reasoning
... your reasoning for what matters and how to optimize it ...
# Refined Observation
... reduced and actionable observation ...
```

**Prompt E.9: AppWorld Prompt for observation compression after optimization (UT)**

```
Your task: write two sections---"Reasoning" and "Refined
    Observation".

1. Reasoning
   - Examine task, history, and observation.
   - Decide exactly which parts of the observation must be kept
    so the next agent step can succeed.
   - Note any need to paginate (page_limit default = 5,
    page_index).
   - Justify any data you drop.

2. Refined Observation
```

```
       - Contain only the minimal yet sufficient info for the next
        step.
      - Always preserve:
        - Every endpoint that may be called, plus its full
       parameter list and defaults (especially page_limit/
       page_index, auth tokens).
        - Response-schema fields referenced or likely needed later
       (e.g., play_count, release_date, like_count, position, ids).
        - Raw data rows required for future comparisons or loops;
       if summarising, keep at least all positive-match examples.
      - Never:
        - Omit defaults that affect behaviour.
        - Declare parameters "not critical" without proof.
        - Hallucinate endpoints or fields.
        - Replace machine-readable data with vague prose.

  [Information source]
  # User Instruction
  {{ task }}

  # History of interactions
  {{ history }}

  # Observation at the current time step
  {{ observation }}

  [Output format]
  # Reasoning
  ...concise analysis explaining what is kept/removed...

  # Refined Observation
  ...trimmed yet complete observation ensuring future steps remain
      possible...
```

**Prompt E.10: AppWorld Prompt for observation compression after optimization (UTCO)**

```
Your task: create two sections---"Reasoning" and "Refined
   Observation".

1. Reasoning (<=40 words)
   - Briefly state what was kept and why; note dropped
    categories and any pagination needs.

2. Refined Observation (use ONLY the current observation)
   - Keep strictly necessary data for the next step.
   - Always preserve, when present:
     - access_token or other auth values
     - page_limit, page_index and other defaults that affect
    calls
     - every endpoint name; include parameters only if required;
    description <= 4 words
     - identifiers/fields needed for comparisons, loops, or API
    calls (ids, titles, counts, paths, etc.)
   - Minimise length:
     - Delete unused fields and narrative text.
     - Compress JSON/arrays (one object per line, no extra
    spaces).
     - Summarise long uniform lists with a range/pattern when
    individual rows are not needed.
   - Never invent, alter, or omit a required literal.
```

```
   - Exclude history, prior summaries, and meta comments.

[Information source]
# User Instruction
{{ task }}

# History of interactions
{{ history }}

# Observation at the current time step
{{ observation }}

[Output format]
# Reasoning
...

# Refined Observation
...
```

**Example E.1: AppWorld task trajectory example**

**Problem:** Name the artist least recommended to me on Spotify.

```python
# First, let's check what APIs are available under the Spotify app to
↪  see if there is an API related to recommendations or artists.
print(apis.api_docs.show_api_descriptions(app_name='spotify'))
```

```
[{"name":"login",...},{"name":"show_recommendations","descriptio
↪  n":"Get personalized song recommendations for the user."},
↪  ...]
```

```python
# There is an API called "show_recommendations" which provides
↪  personalized song recommendations. Let's check its detailed
↪  specification...
print(apis.api_docs.show_api_doc(app_name='spotify',
↪  api_name='show_recommendations'))
```

```
{"path":"/recommendations","method":"GET","parameters":[{"name":
↪  "access_token","required":true},
↪  {"name":"page_index",...},{"name":"page_limit",...}], ...}
```

```python
# To call `show_recommendations`, I need an access token. Check the
↪  login spec.
print(apis.api_docs.show_api_doc(app_name='spotify',
↪  api_name='login'))
```

```
{"path":"/auth/token","method":"POST","parameters":[{"name":"use
↪  rname"},{"name":"password"}],
↪  ...}
```

```python
# I need Timothy's Spotify account password to log in. Let's get it
↪  from supervisor.
spotify_passwords = apis.supervisor.show_account_passwords()
spotify_password = next((acc["password"] for acc in spotify_passwords
↪  if acc["account_name"]=="spotify"), None)
print(spotify_password)
```

```
<REDACTED>
```

```python
# Now log in to get the access token.
spotify_login = apis.spotify.login(username="timothy.whit@gmail.com",
                                   password="<REDACTED>")
access_token = spotify_login.get("access_token")
print(access_token)
```

```
eyJhbGciOiJIUzI1NiIsInR5cCI6IkpXVCJ9...<REDACTED>...
```

```python
# Now get the first page of personalized song recommendations.
recommendations_page_0 = apis.spotify.show_recommendations(
    access_token=access_token, page_index=0, page_limit=20
)
print(recommendations_page_0[:4])  # show a few items only
```

```
[
 {"song_id":10,"title":"The Curse of Loving
 ↪  You","artists":[{"name":"Lucas Grey"}]},
 {"song_id":11,"title":"Lost in a Moment's
 ↪  Grace","artists":[{"name":"Apollo
 ↪  Serenade"},{"name":"Phoenix Rivers"}]},
 {"song_id":40,"title":"Burning Like a Distant
 ↪  Star","artists":[{"name":"Aria Sterling"}]},
 ...
]
```

```
# To find the artist least recommended to Timothy, I will count how
↪  many times each artist appears in the recommendations. The artist
↪  with the fewest appearances is the least recommended.
from collections import Counter

artist_names = [a["name"] for r in recommendations_page_0 for a in
↪  r["artists"]]
artist_counts = Counter(artist_names)
least_recommended_artist = min(artist_counts, key=artist_counts.get)
print(least_recommended_artist)
```

```
  Evelyn Rose
```

```
# Task complete. I will submit "Evelyn Rose" as the answer.
apis.supervisor.complete_task(answer="Evelyn Rose")
```

**Example E.2: AppWorld trajectory example - gpt-4.1-mini agent fails without history compression**

**Problem:** Delete all `.pdf` files from `~/downloads`.

```
# Understand available APIs in the file_system app.

print(apis.api_docs.show_api_descriptions(app_name='file_system'))
```

```
[
{"name":"show_account","description":"Show your account
↪  information. Unlike show_profile, this includes private
↪  information."},
{"name":"signup","description":"Sign up to create account."},
{"name":"delete_account","description":"Delete your account."},
{"name":"update_account_name","description":"Update your first or
↪  last name in the account profile."},
...
{"name":"copy_directory","description":"Copy a directory to
↪  another location."},
{"name":"move_directory","description":"Move a directory to
↪  another location."},
{"name":"compress_directory","description":"Compress a
↪  directory."},
{"name":"decompress_file","description":"Decompress a compressed
↪  file."}
]
```

```
# List files in ~/downloads to find PDFs.
```

```
2106
2107   downloads_path = '~/downloads'
2108   downloads_files = apis.file_system.show_directory(path=downloads_path)
       print(downloads_files)
2109
2110
         Execution failed. Traceback:
2111
         ...
2112     Exception: Response status code is 401:
2113     {"message":"You are either not authorized to access this
2114   ↪   file_system API endpoint or your access token is missing,
       ↪   invalid or expired."}
2115
2116
2117
2118   # Need authentication. Check supervisor for stored credentials.
2119
       print(apis.supervisor.show_account_passwords())
2120
2121
         [
2122     {"account_name":"amazon","password":"Gt$!_*W"},
2123     {"account_name":"file_system","password":"8nNw!jZ"},
2124     {"account_name":"gmail","password":"qu4Y7}s"},
2125     {"account_name":"phone","password":"C4n&I40"},
         {"account_name":"simple_note","password":"e+QwbmV"},
2126     {"account_name":"splitwise","password":"mSqG}QU"},
2127     {"account_name":"spotify","password":"%CCvl8v"},
2128     {"account_name":"todoist","password":"jHZ#RPM"},
2129     {"account_name":"venmo","password":"Wq8!RAU"}
         ]
2130
2131
2132
2133   # Try logging in with just the password (no username).
2134
       file_system_password = [a["password"] for a in
2135   ↪   apis.supervisor.show_account_passwords()
2136   if a["account_name"] == "file_system"][0]
       file_system_login_response =
2137   ↪   apis.file_system.login(password=file_system_password)
2138   print(file_system_login_response)
2139
2140
         Execution failed. Traceback:
2141     ...
2142     Exception: Response status code is 422:
2143     {"message":"Validation error. Reason: \nusername: Field
2144   ↪   required"}
2145
2146
2147   # Need username. Check supervisor profile for email/username.
2148
       print(apis.supervisor.show_profile())
2149
2150
         {"first_name":"Susan","last_name":"Burton","email":"[susanmiller
2151   ↪   @gmail.com](mailto:susanmiller@gmail.com)",
2152     "phone_number":"3296062648","birthday":"1994-04-30","sex":"femal
2153   ↪   e"}
2154
2155
2156   # Attempt login with email as username.
2157
       file_system_username =
2158   ↪   "[susanmiller@gmail.com](mailto:susanmiller@gmail.com)"
2159
```

```
file_system_password = [a["password"] for a in
↪  apis.supervisor.show_account_passwords()
if a["account_name"] == "file_system"][0]
file_system_login_response = apis.file_system.login(
username=file_system_username, password=file_system_password)
print(file_system_login_response)
```

```
{"access_token":"eyJhbGciOiJIUzI1NiIsInR5cCI6IkpXVCJ9...<REDACTE↓
↪  D>...","token_type":"Bearer"}
```

```
# Try to set token manually (guessing an API), then list directory.

apis.file_system.set_access_token("...<REDACTED>...")
downloads_path = '~/downloads'
downloads_files = apis.file_system.show_directory(path=downloads_path)
print(downloads_files)
```

```
Execution failed. Traceback:
...
Exception: No API named 'set_access_token' found in the
↪  file_system app.
```

```
# Call show_directory again, assuming session is persisted after
↪  login.

downloads_path = '~/downloads'
downloads_files = apis.file_system.show_directory(path=downloads_path)
print(downloads_files)
```

```
Execution failed. Traceback:
...
Exception: Response status code is 401:
{"message":"You are either not authorized to access this
↪  file_system API endpoint or your access token is missing,
↪  invalid or expired."}
```

```
# Retry: fresh login, try immediate call.

login_response = apis.file_system.login(
username=file_system_username, password=file_system_password)
access_token = login_response.get("access_token")
print(access_token)  # for debugging
```

```
eyJhbGciOiJIUzI1NiIsInR5cCI6IkpXVCJ9...<REDACTED>...
```

```
# Try show_directory once more after login.

downloads_path = '~/downloads'
downloads_files = apis.file_system.show_directory(path=downloads_path)
print(downloads_files)
```

```
  Execution failed. Traceback:
  ...
  Exception: Response status code is 401:
  {"message":"You are either not authorized to access this
  ↪  file_system API endpoint or your access token is missing,
  ↪  invalid or expired."}
```

```python
# Inspect file_system APIs again for any token/session helpers.

print(apis.api_docs.show_api_descriptions(app_name='file_system'))
```

```
  [{"name":"show_account","description":"Show your account
  ↪  information. Unlike show_profile, this includes private
  ↪  information."},
  {"name":"signup","description":"Sign up to create account."},
  {"name":"delete_account","description":"Delete your account."},
  {"name":"update_account_name","description":"Update your first or
  ↪  last name in the account profile."},
  {"name":"login","description":"Login to your account."},
  ...
  {"name":"copy_directory","description":"Copy a directory to
  ↪  another location."},
  {"name":"move_directory","description":"Move a directory to
  ↪  another location."},
  {"name":"compress_directory","description":"Compress a
  ↪  directory."},
  {"name":"decompress_file","description":"Decompress a compressed
  ↪  file."}]
```

```python
# Try logout -> login -> call again.

apis.file_system.logout()
login_response = apis.file_system.login(
username=file_system_username, password=file_system_password)
downloads_path = '~/downloads'
downloads_files = apis.file_system.show_directory(path=downloads_path)
print(downloads_files)
```

```
  Execution failed. Traceback:
  ...
  Exception: Response status code is 422:
  {"message":"You are either not authorized to access this
  ↪  file_system API endpoint or your access token is missing,
  ↪  invalid or expired."}
```

```python
# Final retry without logout; still unauthorized.

login_response = apis.file_system.login(
username=file_system_username, password=file_system_password)
downloads_path = '~/downloads'
downloads_files = apis.file_system.show_directory(path=downloads_path)
print(downloads_files)
```

```
  Execution failed. Traceback:
  ...
  Exception: Response status code is 401:
  {"message":"You are either not authorized to access this
  ↪  file_system API endpoint or your access token is missing,
  ↪  invalid or expired."}
```

```
# Supervisor cannot manage files; mark task failed due to
↪  authorization issues.

print(apis.api_docs.show_api_descriptions(app_name='supervisor'))
```

```
  [
  {"name":"show_active_task","description":"Show the currently
  ↪  active task assigned to you by the supervisor."},
  {"name":"complete_task","description":"Mark the currently active
  ↪  task as complete with the given answer."},
  {"name":"show_profile","description":"Show your supervisor's
  ↪  profile information."},
  {"name":"show_addresses","description":"Show your supervisor's
  ↪  addresses."},
  {"name":"show_payment_cards","description":"Show your
  ↪  supervisor's payment_cards."},
  {"name":"show_account_passwords","description":"Show your
  ↪  supervisor's app account passwords."}
  ]
```

```
apis.supervisor.complete_task(status="fail")
```

**Example E.3:** AppWorld trajectory example - gpt-4.1-mini agent successes with history compression

**Problem:** Delete all `.pdf` files from `~/downloads`.

```
(... previous raw history is omitted for brevity ...)
<HISTORY_SUMMARY>

1. REASONING
   - User requested deletion of all .pdf files in ~/downloads.
   - file_system APIs require authentication; initial calls
   ↪   failed with 401 Unauthorized.
   (... omitted ...)
   - Confirmed login API returns access_token but no mechanism to
   ↪   use it in subsequent calls found.
   - Need to find a way to authorize file_system API calls with
   ↪   the access token or alternative approach.

2. VARS
   | name                  | value
   ↪   | purpose                   |
   |---------|----------|--------|
   | file_system_username | "susanmiller@gmail.com"
   ↪   | Username for file_system login  |
   | file_system_password | "8nNw!jZ"
   ↪   | Password for file_system login  |
   | file_system_token    |
   ↪   "eyJhbGciOiJIUzI1NiIsInR5cCI6IkpXVCJ9...sBA8nTOB..." |
   ↪   Access token from login |
   | downloads_path       | "~/downloads"
   ↪   | Target directory for file listing |

3. TODO
   - Determine how to pass or set the access token for
   ↪   file_system API calls to authorize them.
   - Once authorized, list files in ~/downloads to identify .pdf
   ↪   files.
   - Delete all .pdf files found in ~/downloads.

4. COMPLETED
   - Retrieved file_system account password from supervisor app.
   - Successfully logged in to file_system app with username and
   ↪   password, obtained access token.
   - Confirmed file_system APIs require authorization; calls
   ↪   without token fail.
   - Verified no supervisor app API to proxy file system
   ↪   operations.
   - Confirmed login API returns access token but no visible
   ↪   method to use it in subsequent calls.

5. GUARDRAILS
   - Always provide both username and password for login API.
   - Access tokens must be explicitly handled or passed if
   ↪   environment does not auto-use them.
   - Validate authorization before calling protected APIs to
   ↪   avoid 401 errors.
   - Avoid redundant login attempts without token usage
   ↪   resolution.
   - Preserve tokens and credentials securely for session
   ↪   continuity.
   </HISTORY_SUMMARY>

# The show_directory API requires the access_token as a parameter.
```

```python
# List files in "~/downloads" using the token.

downloads_path = "~/downloads"
file_system_access_token =
↪   "eyJhbGciOiJIUzI1NiIsInR5cCI6IkpXVCJ9...sBA8nTOB..."

print(apis.file_system.show_directory(
access_token=file_system_access_token,
directory_path=downloads_path,
entry_type="files",
recursive=False))
```

```
  [
  "/home/susan/downloads/aerobics_class.jpg",
  "/home/susan/downloads/art_inspiration_sketches.zip",
  "/home/susan/downloads/bakeware.pdf",
  "/home/susan/downloads/bowls_for_snacks.pdf",
  "/home/susan/downloads/candy.pdf",
  "/home/susan/downloads/charcoal_pencils.pdf",
  ...,
  "/home/susan/downloads/travel_insurance.pdf",
  "/home/susan/downloads/workout_clothes.pdf",
  "/home/susan/downloads/world_cuisine_recipe_book.epub",
  "/home/susan/downloads/world_landmarks_photo_album.zip"
  ]
```

```python
# Filter to only ".pdf" files.

downloads_files = [
"/home/susan/downloads/aerobics_class.jpg",
"/home/susan/downloads/bakeware.pdf",
"/home/susan/downloads/bowls_for_snacks.pdf",
"/home/susan/downloads/candy.pdf",
"/home/susan/downloads/charcoal_pencils.pdf",
...,
"/home/susan/downloads/travel_insurance.pdf",
"/home/susan/downloads/workout_clothes.pdf"
]
pdf_files = [p for p in downloads_files if p.lower().endswith(".pdf")]
print(pdf_files)
```

```
  [
  "/home/susan/downloads/bakeware.pdf",
  "/home/susan/downloads/bowls_for_snacks.pdf",
  "/home/susan/downloads/candy.pdf",
  "/home/susan/downloads/charcoal_pencils.pdf",
  ...,
  "/home/susan/downloads/travel_insurance.pdf",
  "/home/susan/downloads/workout_clothes.pdf"
  ]
```

```python
# Confirm delete_file API parameters before deletion.

print(apis.api_docs.show_api_doc(app_name="file_system",
↪   api_name="delete_file"))
```

```
    {
    "app_name":"file_system",
    "api_name":"delete_file",
    "path":"/file",
    "method":"DELETE",
    "description":"Delete a file.",
    "parameters":[
    {"name":"file_path","type":"string","required":true,
    "description":"Path of the file. Path can be absolute, starting
    ↪   with '/', or relative to the user's home directory, starting
    ↪   with '~/'."},
    {"name":"access_token","type":"string","required":true,
    "description":"Access token obtained from file_system app
    ↪   login."}
    ],
    "response_schemas":{
    "success":{"message":"string","file_path":"string"},
    "failure":{"message":"string"}
    }
    }
```

```python
# Delete each PDF using the access_token.

delete_results = []
for pdf in pdf_files:
res = apis.file_system.delete_file(
file_path=pdf, access_token=file_system_access_token)
delete_results.append(res)

# Mark task complete.

apis.supervisor.complete_task()
```

## Compression E.1: AppWorld History Compression Example

```
task: I got some Splitwise group invitations over phone text
    messages yesterday.
If their number is in my phone contact book, accept it, otherwise
    delete those messages.

----------------------------------------

History Compression (Prompting baseline):
### REASONING

The agent began by identifying relevant apps for handling text
    messages and Splitwise invitations, then explored the phone app
    's APIs to manage messages and contacts. To search for
    Splitwise group invitations received via text message, the
    agent determined it needed to log in to the phone app to obtain
     an access token. Using credentials from the supervisor app,
    the agent successfully logged in and retrieved the access token
    . The agent then calculated yesterday's date range and searched
     for text messages containing "Splitwise," retrieving a list of
     messages. The rationale for retaining message details (sender
    name, phone number, contact_id, message content, sent_at) is to
     enable the next step: checking if the sender's number is in
    the contact book, so the agent can accept or delete invitations
     as instructed. Earlier steps (API exploration, login, date
    calculation) directly enabled the message search and data
    extraction.

### COMPLETED

- Identified relevant apps and APIs for the task.
- Successfully logged in to the phone app and obtained access token
    .
- Calculated yesterday's date range.
- Searched and retrieved text messages mentioning "Splitwise" from
    yesterday and beyond, including sender details and message
    content.

----------------------------------------

History Compression (ACON (utility step)):
1. REASONING
   - Identified "phone" app for managing text messages and contacts
    .
   - Retrieved API docs to find endpoints for searching messages
    and contacts.
   - Determined yesterday's date (May 17, 2023) for filtering
    relevant messages.
   - Checked "search_text_messages" API spec for required
    parameters.
   - Used supervisor app to obtain phone app password.
   - Logged in to phone app to get access_token.
   - Searched for messages containing "Splitwise" with page_limit
    =20, sorted by -created_at.
   - Located Splitwise group invitation messages; sender contact_id
     used to determine if sender is in contact book.

2. VARS
   | name          | value
                            | purpose
       |
   |--------------|--------------------------|------------------|
```

```
    | access_token  | eyJhbGciOiJIUzI1NiIsInR5cCI6IkpXVCJ9.
    eyJzdWIiOiJwaG9uZSs3OTM0ODE4MTkxIiwiZXhwIjoxNjg0NDEyMDk4fQ.
    ad3t3S_Xj7YhoDDFZeW4BlVL4dNniMdfaXC1143fbzw | Auth for phone
    app APIs                    |
    | phone_number  | 7934818191
                            | User's phone number
      |
    | phone_password| X53y6aW
                              | Phone app password
      |
    | search_query  | "Splitwise"
                            | Filter for relevant messages
        |
    | sort_by       | -created_at
                              | Get latest messages first
        |
    | page_index    | 0
                              | Current page index for pagination
        |
    | page_limit    | 20
                              | Number of results per page
        |
    | target_date   | 2023-05-17
                            | Date to filter messages
        |
    | invitation_msgs | [IDs: 16794, 16800]
                            | Splitwise invitations from 2023-05-17
        |
    | sender_info   | [{id:1857, name:"Alex White", phone
    :"3707865891"}, {id:null, name:"Anita Burch", phone
    :"3643463570"}] | For contact book check                   |

3. TODO
   - For each Splitwise invitation message from 2023-05-17:
     - Check if sender's phone number exists in contact book (
   search_contacts API).
     - If in contact book, accept invitation (Splitwise API, if
   needed).
     - If not in contact book, delete the message (
   delete_text_message API).
   - Paginate search_text_messages if more relevant messages may
   exist.

4. COMPLETED
   - Identified relevant apps and APIs for task.
   - Retrieved phone app password from supervisor.
   - Logged in to phone app and obtained access_token.
   - Searched and filtered text messages for Splitwise invitations
   on target date.
   - Isolated relevant message IDs and sender info for next steps.

5. GUARDRAILS
   - Always re-declare VARS after memory reset.
   - Paginate message search until no more results.
   - Validate API parameters before each call.
   - Avoid redundant logins and API doc look-ups.
   - Only act on messages matching both query and target date.

----------------------------------------

History Compression (ACON (utility step + compression step)):
```

```
1. REASONING
   I identified voice messages from the day before yesterday
    (2023-05-16) containing Splitwise group invitations. To follow
    instructions, I need to check if the senders' phone numbers are
     in Sierra's contact book before accepting or deleting these
    messages.

2. VARS
   access_token=<token>, splitwise_invite_voice_ids
    =[17022,17036,17028,17034], sender_numbers
    =['9595488531','7375474944','2123782445','3808719492']

3. TODO
   - Search Sierra's contact book for sender_numbers.
   - Accept Splitwise invitations from contacts found.
   - Delete voice messages from non-contacts.

4. COMPLETED
   - Logged in to phone app and obtained access_token.
   - Retrieved voice messages from 2023-05-16.
   - Identified Splitwise invitation messages and sender numbers.

5. GUARDRAILS
   Only process Splitwise invitations from contacts; delete others.
```

## Compression E.2: 8-objective QA History Compression Example

```
task: when was the last time the dodgers played yankees in the
    world series?; where will be the next olympics be held?; who is
     tinker air force base named after?; who does the voice of cruz
     on cars 3?; who's the original singer of help me make it
    through the night?; where does the movie mothers day take place
    ?; what is the big gold dome in jerusalem?; when did russia
    join the world economic forum?

--------------------------------------

History Compression (Prompting baseline):
### REASONING
The agent began by addressing the user's multi-part query, starting
     with "when was the last time the Dodgers played Yankees in the
     World Series?" Through Wikipedia searches, the agent
    identified 1981 as the most recent World Series meeting between
     the Dodgers and Yankees. This answer was retained due to clear
     evidence in the retrieved documents. The agent then attempted
    to answer "where will be the next Olympics be held?" but
    initial searches returned irrelevant or outdated results (e.g.,
     Amsterdam 1928, Buenos Aires 2018). Recognizing the lack of
    direct answers, the agent refined its search queries multiple
    times, but still did not retrieve the current location for the
    next Olympics. The agent's iterative approach demonstrates a
    commitment to accuracy and highlights the need to adjust search
     strategies when initial attempts fail. Retaining details of
    failed searches helps prevent repeated mistakes and guides
    future queries.

### COMPLETED
- Identified the last time the Dodgers played the Yankees in the
    World Series: 1981.

--------------------------------------
```

```
History Compression (ACON (utility step)):

### REASONING
- Used Wikipedia search for each question; refined Olympics query
    after initial results lacked direct info.
- Collapsed repeated/irrelevant Olympic bid results; focused on
    confirmed host city.
- For Tinker AFB, confirmed name origin from first document.

### COMPLETED
- Last Dodgers vs Yankees World Series: 1955 (source: "1955 World
    Series")
- Next Olympics location: Paris, 2024 (source: "Budapest bid for
    the 2024 Summer Olympics")
- Tinker Air Force Base named after Major General Clarence L.
    Tinker (source: "Tinker Air Force Base")

### STATE LOG
FACT CACHE:
- When was the last time the Dodgers played Yankees in the World
    Series? -> 1955 ("1955 World Series")
- Where will the next Olympics be held? -> Paris, 2024 ("Budapest
    bid for the 2024 Summer Olympics")
- Who is Tinker Air Force Base named after? -> Major General
    Clarence L. Tinker ("Tinker Air Force Base")

PENDING:
- Who does the voice of Cruz on Cars 3?
- Who is the original singer of "Help Me Make It Through the Night
    "?
- Where does the movie "Mother's Day" take place?
- What is the big gold dome in Jerusalem?
- When did Russia join the World Economic Forum?

ATTEMPTED QUERIES / LOOP GUARD:
- "Dodgers Yankees World Series history" OK
- "next Olympics location" FAIL (no direct info)
- "2024 Olympics location" OK
- "Tinker Air Force Base name origin" OK
- "next Olympics location" tried twice (loop guard: must
    reformulate/abandon)

ERROR NOTES:
None

(End of summary - no extra text)

-----------------------------------------

History Compression (ACON (utility step + compression step)):

### FACT CACHE
- When was the last time the Dodgers played Yankees in the World
    Series? -> 1956

### PENDING
where will the next olympics be held?, who is tinker air force base
    named after?, who does the voice of cruz on cars 3?, who is
    the original singer of help me make it through the night?,
    where does the movie mothers day take place?, what is the big
    gold dome in jerusalem?, when did russia join the world
    economic forum?
```

```
### BLOCKED
next Olympics location: 3x failed searches (loop-guard)

### REASONING
Olympics location queries returned irrelevant results; alternate
    sources or query phrasing needed.
```

---

**Compression E.3: OfficeBench History Compression Example**

```
task: read company revenues, save the top 2 revenue years in top2.
    xlsx, generate reports.docx

----------------------------------------

History Compression (Prompting baseline):
### REASONING

The agent began by identifying the relevant data file (
    company_revenues.xlsx) containing company revenue information.
    After reading the file, the agent determined the top 2 revenue
    years: 2014 ($4,983,791) and 2015 ($4,611,210). The agent then
    created a new Excel file (top2.xlsx) and started populating it
    with the required data, first adding headers ("Year", "Revenue
    ($)") and then entering the first top year (2014) in the second
     row, first column. The rationale for retaining row and column
    indices, file paths, and the top 2 years is to ensure accurate
    and complete transfer of data and to facilitate the next steps
    (completing the top2.xlsx file and generating reports.docx).

### COMPLETED

- Located and read company_revenues.xlsx.
- Identified top 2 revenue years: 2014 ($4,983,791), 2015 ($4
    ,611,210).
- Created top2.xlsx and added headers ("Year", "Revenue ($)").
- Entered "2014" in (2, 1) of top2.xlsx.

----------------------------------------

History Compression (ACON (utility step)):

### REASONING
- Located company revenue data in company_revenues.xlsx.
- Read all rows; identified top 2 revenue years: 2014 ($4,983,791)
    and 2015 ($4,611,210).
- Created top2.xlsx and began writing header and first top year.
- Each numeric value was copied verbatim from source.
- No errors or mismatches so far; headers and first year written
    successfully.
- Pending: finish writing revenue for 2014, add 2015 row, then
    generate reports.docx.

Verify numeric values directly from source file before use.

---

### DATA STATE

#### company_revenues.xlsx (source table)
| Row | Year | Revenue ($) |
|-----|------|-------------|
| 2   | 2004 | 4439044     |
```

```
| 3   | 2005 | 2212203      |
| 4   | 2006 | 1193225      |
| 5   | 2007 | 2793265      |
| 6   | 2008 | 2080460      |
| 7   | 2009 | 1791112      |
| 8   | 2010 | 3098049      |
| 9   | 2011 | 3975966      |
| 10  | 2012 | 3251777      |
| 11  | 2013 | 2918667      |
| 12  | 2014 | 4983791      |
| 13  | 2015 | 4611210      |
| 14  | 2016 | 3294631      |
| 15  | 2017 | 1463500      |
| 16  | 2018 | 3945846      |
| 17  | 2019 | 3907271      |
| 18  | 2020 | 3651262      |
| 19  | 2021 | 2005820      |
| 20  | 2022 | 2170828      |
| 21  | 2023 | 2832545      |

#### top2.xlsx (current state)
| (row, col) | Value         |
|------------|---------------|
| (1, 1)     | Year          |
| (1, 2)     | Revenue ($)   |
| (2, 1)     | 2014          |
| (2, 2)     | [Empty Cell]  |

---

### TODO / PENDING
- Write 2014 revenue (4983791) to (2,2) in top2.xlsx.
- Write 2015 year (3,1) and revenue (4611210) (3,2) in top2.xlsx.
- Generate reports.docx summarizing top 2 revenue years.
- Re-read source rows before writing each numeric value for
    consistency.

---

### COMPLETED
- Located and read company_revenues.xlsx.
- Created top2.xlsx.
- Wrote headers and 2014 year to top2.xlsx.
```

---

**Compression E.4: AppWorld Observation Compression Example**

```
task: I am going for a 15-minute drive without internet. Play an
    album from my Spotify library that already has enough
    downloaded songs for it, so I do not have to repeat.

action: print(apis.api_docs.show_api_descriptions(app_name='spotify
    '))

----------------------------------------

Raw Observation:
[
  {
   "name": "show_account",
   "description": "Show your account information. Unlike
    show_profile, this includes private information."
  },
```

```
{
 "name": "signup",
 "description": "Sign up to create account."
},
{
 "name": "delete_account",
 "description": "Delete your account."
},
{
 "name": "update_account_name",
 "description": "Update your first or last name in the account
   profile."
},
{
 "name": "login",
 "description": "Login to your account."
},
{
 "name": "logout",
 "description": "Logout from your account."
},
{
 "name": "send_verification_code",
 "description": "Send account verification code to your email
   address."
},
{
 "name": "verify_account",
 "description": "Verify your account using the verification code
   sent to your email address."
},
{
 "name": "send_password_reset_code",
 "description": "Send password reset code to your email address."
},
{
 "name": "reset_password",
 "description": "Reset your password using the password reset code
   sent to your email address."
},
{
 "name": "show_profile",
 "description": "Show public profile information of a user."
},
{
 "name": "show_genres",
 "description": "Show the list of all music genres."
},
{
 "name": "search_songs",
 "description": "Search for songs with a query."
},
{
 "name": "show_song",
 "description": "Get details of a specific song."
},
{
 "name": "show_song_privates",
 "description": "Show information about the song that is private
   to the user."
},
{
 "name": "like_song",
```

```
    "description": "Like a song."
   },
   {
    "name": "unlike_song",
    "description": "Unlike a song."
   },
   {
    "name": "show_liked_songs",
    "description": "Get a list of songs you have liked."
   },
   {
    "name": "search_albums",
    "description": "Search for albums with a query."
   },
   {
    "name": "show_album",
    "description": "Get details of a specific album."
   },
   {
    "name": "show_album_privates",
    "description": "Show information about the album that is private
     to the user."
   },
   {
    "name": "like_album",
    "description": "Like a album."
   },
   {
    "name": "unlike_album",
    "description": "Unlike a album."
   },
   {
    "name": "show_liked_albums",
    "description": "Get a list of albums you have liked."
   },
   {
    "name": "show_playlist_library",
    "description": "Get a list of playlists in the user's playlist
     library."
   },
   {
    "name": "search_playlists",
    "description": "Search for playlists with a query. It will search
     over all public playlists and your own private playlists."
   },
   {
    "name": "create_playlist",
    "description": "Create a new playlist."
   },
   {
    "name": "show_playlist",
    "description": "Get detailed information about a specific
     playlist. You can view your own playlists or others' playlists
     if they are public."
   },
   {
    "name": "delete_playlist",
    "description": "Delete a playlist."
   },
   {
    "name": "update_playlist",
    "description": "Update a playlist title or privacy."
   },
```

```
  {
   "name": "show_playlist_privates",
   "description": "Show information about the playlist that is
     private to the user."
  },
  {
   "name": "like_playlist",
   "description": "Like a playlist."
  },
  {
   "name": "unlike_playlist",
   "description": "Unlike a playlist."
  },
  {
   "name": "show_liked_playlists",
   "description": "Get a list of playlists you have liked."
  },
  {
   "name": "search_artists",
   "description": "Search for artists with a query."
  },
  {
   "name": "show_artist",
   "description": "Get details of a specific artist."
  },
  {
   "name": "show_artist_following",
   "description": "Show if the user is following the artist."
  },
  {
   "name": "show_song_library",
   "description": "Get a list of songs in the user's song library."
  },
  {
   "name": "add_song_to_library",
   "description": "Add a song to the user's song library."
  },
  {
   "name": "remove_song_from_library",
   "description": "Remove a song from the user's song library."
  },
  {
   "name": "show_album_library",
   "description": "Get a list of albums in the user's album library
     ."
  },
  {
   "name": "add_album_to_library",
   "description": "Add an album to the user's album library."
  },
  {
   "name": "remove_album_from_library",
   "description": "Remove an album from the user's album library."
  },
  {
   "name": "add_song_to_playlist",
   "description": "Add a song to a playlist."
  },
  {
   "name": "remove_song_from_playlist",
   "description": "Remove a song from a playlist."
  },
  {
```

```
 "name": "show_downloaded_songs",
 "description": "Get a list of downloaded songs."
},
{
 "name": "download_song",
 "description": "Download a song."
},
{
 "name": "remove_downloaded_song",
 "description": "Remove a song from downloads."
},
{
 "name": "show_following_artists",
 "description": "Get a list of artists the user is following."
},
{
 "name": "follow_artist",
 "description": "Follow an artist."
},
{
 "name": "unfollow_artist",
 "description": "Unfollow an artist."
},
{
 "name": "show_song_reviews",
 "description": "Get a list of reviews for a song."
},
{
 "name": "review_song",
 "description": "Rate or review a song."
},
{
 "name": "show_song_review",
 "description": "Show a song review."
},
{
 "name": "delete_song_review",
 "description": "Delete a song review."
},
{
 "name": "update_song_review",
 "description": "Update a song review."
},
{
 "name": "show_album_reviews",
 "description": "Get a list of reviews for an album."
},
{
 "name": "review_album",
 "description": "Rate or review an album."
},
{
 "name": "show_album_review",
 "description": "Show an album review."
},
{
 "name": "delete_album_review",
 "description": "Delete an album review."
},
{
 "name": "update_album_review",
 "description": "Update an album review."
},
```

```
{
 "name": "show_playlist_reviews",
 "description": "Show a list of reviews for your playlist or
   others' public playlist."
},
{
 "name": "review_playlist",
 "description": "Rate or review a playlist."
},
{
 "name": "show_playlist_review",
 "description": "Show a playlist review."
},
{
 "name": "delete_playlist_review",
 "description": "Delete a playlist review."
},
{
 "name": "update_playlist_review",
 "description": "Update a playlist review."
},
{
 "name": "show_payment_cards",
 "description": "Get a list of users payment cards."
},
{
 "name": "add_payment_card",
 "description": "Add a new payment card."
},
{
 "name": "show_payment_card",
 "description": "Get details of a payment card."
},
{
 "name": "delete_payment_card",
 "description": "Delete payment card information."
},
{
 "name": "update_payment_card",
 "description": "Update payment card information."
},
{
 "name": "show_current_song",
 "description": "Show details of the current song on the queue."
},
{
 "name": "play_music",
 "description": "Play music based on various criteria. You can
   pass, at most, any one of queue_position, song_id, album_id or
   playlist_id. If one of song_id, album_id or playlist_id is
   passed, that song, album or playlist will be added to the queue
    and played. Otherwise, the queue will remain unchanged. If
   queue_position is passed, the song at that position in the
   queue will be played. If none is passed, the current song in
   the queue will be played."
},
{
 "name": "pause_music",
 "description": "Pause the currently playing song."
},
{
 "name": "previous_song",
 "description": "Go to the previous song in the song queue."
```

```
        },
        {
         "name": "next_song",
         "description": "Go to the next song in the song queue."
        },
        {
         "name": "move_song_in_queue",
         "description": "Move a song in the queue to a new position."
        },
        {
         "name": "seek_song",
         "description": "Seek the current song to the given number of
           seconds."
        },
        {
         "name": "loop_song",
         "description": "Set whether to loop the current song."
        },
        {
         "name": "shuffle_song_queue",
         "description": "Shuffle songs in the music player queue."
        },
        {
         "name": "show_song_queue",
         "description": "Get the music player song queue. Songs are played
           in the order of the queue in a cycle."
        },
        {
         "name": "add_to_queue",
         "description": "Add a song, album or playlist to the music player
           song queue."
        },
        {
         "name": "clear_song_queue",
         "description": "Clear the music player song queue."
        },
        {
         "name": "remove_song_from_queue",
         "description": "Remove a song at the given position from the
          music player song queue."
        },
        {
         "name": "show_volume",
         "description": "Get the volume level of the music player."
        },
        {
         "name": "set_volume",
         "description": "Set the volume level of the music player."
        },
        {
         "name": "show_recommendations",
         "description": "Get personalized song recommendations for the
           user."
        },
        {
         "name": "show_premium_plans",
         "description": "Show information about premium plans available."
        },
        {
         "name": "show_premium_subscriptions",
         "description": "Show your premium subscription history."
        },
        {
```

```
   "name": "subscribe_premium",
   "description": "Subscribe to premium membership."
  },
  {
   "name": "download_premium_subscription_receipt",
   "description": "Download the receipt for a premium subscription."
  }
 ]

 ----------------------------------------

 Obs. Compression (Prompting baseline):

 The Spotify API provides:
 - show_album_library: get user's album library.
 - show_downloaded_songs: get list of downloaded songs.
 - show_album: get details of a specific album.

 ----------------------------------------

 Obs. Compression (ACON (utility step)):
 [
   {
     "name": "show_album_library",
     "description": "Get a list of albums in the user's album
     library."
   },
   {
     "name": "show_downloaded_songs",
     "description": "Get a list of downloaded songs."
   },
   {
     "name": "show_album",
     "description": "Get details of a specific album."
   },
   {
     "name": "play_music",
     "description": "Play music based on various criteria. You can
     pass, at most, any one of queue_position, song_id, album_id or
     playlist_id. If one of song_id, album_id or playlist_id is
     passed, that song, album or playlist will be added to the queue
      and played. Otherwise, the queue will remain unchanged. If
     queue_position is passed, the song at that position in the
     queue will be played. If none is passed, the current song in
     the queue will be played."
   }
 ]

 ----------------------------------------

 History Compression (ACON (utility step + compression step)):

 [{"name":"show_album_library","description":"Get user's album
     library."},{"name":"show_downloaded_songs","description":"Get
     downloaded songs."},{"name":"show_album_privates","description
     ":"Show album private info."},{"name":"play_music","description
     ":"Play music; album_id allowed."}]
```

