# OpenReview forum: "ACON: Optimizing Context Compression for Long-horizon LLM Agents"
_ICLR.cc/2026/Conference — Submitted to ICLR 2026_

### Official Review · Reviewer_PhCp · 2025-10-14

**Soundness:** 3
**Presentation:** 4
**Contribution:** 3
**Rating:** 4
**Confidence:** 4

**Summary:**

This paper introduces Agent Context Optimization (ACON), a framework designed to address the challenge of ever-growing context length for LLM agents operating in long-horizon tasks. The core contribution is a novel, gradient-free method for optimizing context compression guidelines. This is achieved by analyzing pairs of trajectories—one with full context that succeeds and one with compressed context that fails—using a powerful LLM to identify the causes of failure and iteratively refine the compression prompt. The authors also propose distilling the optimized compressor into a smaller, more efficient model. Experiments conducted on three benchmarks (AppWorld, OfficeBench, and Multi-objective QA) demonstrate that ACON can significantly reduce peak token usage (26-54%) while largely preserving task performance, and in some cases, even enhancing the capabilities of smaller agent models.

**Strengths:**

Originality and Significance: The paper tackles a highly significant and practical problem for the advancement of LLM agents: context management. The proposed method for optimizing compression guidelines is novel and clever. Using the task outcome (success vs. failure) as a supervisory signal in a gradient-free, natural language optimization loop is an original approach that is broadly applicable, even to closed-source API-based models.


Clarity: The paper is exceptionally well-written and clearly structured. The problem, the proposed solution, and the experimental results are all explained with high clarity. The figures, particularly Figure 1 and 3, are effective at illustrating the core trade-offs and the optimization mechanism.

Empirical Rigor: The experimental evaluation is comprehensive, covering three distinct and challenging long-horizon benchmarks. The results are strong, showing that ACON not only maintains performance close to the "no compression" upper bound but also significantly outperforms other compression baselines, which often suffer from severe performance degradation.

**Weaknesses:**

Practical Cost vs. Token Efficiency: The primary weakness lies in the trade-off between peak token reduction and overall computational/API cost. The paper itself acknowledges this limitation in Section 4.5. While ACON successfully reduces the maximum context length (peak tokens), the process of history compression (which involves frequent calls to the compressor LLM) can break the KV-caching mechanism of the agent LLM. This forces re-computation and can lead to a higher total number of tokens processed and thus higher API costs, as shown in Figure 7. This is a significant practical drawback that might limit the adoption of the history compression part of the framework where cost, not just memory, is the main concern.

Cost and Scalability of the Optimization Process Itself: The paper details the effectiveness of the ACON framework but does not sufficiently discuss the "meta-cost" of the guideline optimization process. This process requires running multiple full trajectories (both with and without compression) and then using a powerful "optimizer" LLM for analysis. For new, complex domains, this optimization phase could be prohibitively expensive and time-consuming. The scalability of this approach to a wide variety of new tasks without incurring substantial upfront costs is unclear.

Generalizability of Optimized Guidelines: The experiments show that guidelines optimized on a specific benchmark's training set work well on its test set. However, the generalizability of these highly specialized guidelines across different domains remains an open question. For instance, would a guideline optimized for AppWorld's application-based tasks be effective for a radically different domain like code generation or scientific literature review? The paper could be strengthened by including an experiment that tests this cross-domain transferability.

**Questions:**

Regarding the critical issue of computational overhead: Could the authors provide a more detailed analysis of the trade-off between peak token reduction and total API cost, especially for history compression? For what types of tasks or interaction patterns does the cost-saving from reduced context outweigh the extra cost of the compressor calls and KV-cache invalidation?

Could the authors elaborate on the cost and complexity of the guideline optimization phase? What is the estimated computational cost (e.g., in terms of number of LLM calls or GPU hours) required to generate an optimized guideline for a new benchmark of similar complexity to AppWorld?

The quality of the optimized guideline seems to depend heavily on the capability of the "optimizer" LLM (O3 model in this case). How sensitive is the final guideline quality to the choice of this optimizer? If a less capable model (e.g., gpt-4.1-mini) were used as the optimizer, would the process still yield significant improvements, or does ACON fundamentally rely on having access to a state-of-the-art reasoning model for optimization?

---

> ### Author Response · Authors · 2025-11-19
> **Response to Reviewer PhCp (1/3)**
>
> We greatly appreciate the reviewer’s thoughtful and insightful comments. It is encouraging to see that our work (1) tackles an important and practical challenge in long-horizon context management, (2) introduces a novel and broadly applicable natural language based optimization method for learning effective compression guidelines, (3) presents the problem, methodology, and results with clear and well structured writing, and (4) delivers strong and comprehensive empirical results across three challenging benchmarks with meaningful gains over existing compression approaches.
>
> To address your concerns, we provide the following clarifications:
> - We clarify that the optimization cost is modest and practical in real settings, and we provide concrete measurements.
> - We clarify that our primary goal is reducing memory cost while preserving performance. We also note that the API cost increase from history compression is minimal and varies across tasks.
> - We clarify that the learned guidelines are intentionally environment-specific, and we explain why this is necessary for strong performance.
> - We clarify the trade-off between peak token reduction and total API cost under different compression thresholds.
> - We provide more details on the dependence on the optimizer model and show that ACON does not rely on a single best model.
>
> Our detailed responses are provided below.
>
> ---
>
> > W1. Practical Cost vs. Token Efficiency.
>
> Thank you for the thoughtful comment. We would like to clarify that our work primarily targets the context explosion problem while keeping inference and API cost low. Our goal is to reduce memory cost, such as peak tokens and dependency as used in previous works [1,2], while preserving or even improving performance under the compressed context.
>
> - **The limited API cost reduction of history compression is not a flaw we overlooked but a limitation we intentionally reported clearly in Section 4.5.** Our intention was to clearly communicate this constraint of LM-based history compression rather than implying that our method provides significant overall API cost reductions.
>     - When using Qwen3-14B as the compressor, API cost increases only slightly on AppWorld ($0.119 to 0.124, about 4 percent), which is a modest overhead.
>     - On 8 objective QA using gpt-4.1-mini as the compressor, API cost decreases by 4%, showing that the cost impact is small and task dependent.
>     - We believe that achieving substantial API cost reductions is an important direction for future work, as discussed in `L923-933`.
> - **Our contributions go well beyond this limitation:**
>     - Guideline optimization restores or improves performance under significant memory cost reduction, achieving up to 54% compression unlike baseline methods (`Tables 1 and 2`).
>     - Distillation enables small compressor models such as Qwen3-14B to match teacher-level compression performance while reducing compressor cost. (`Figure 4`)
>     - Small agents like gpt-4.1-mini and distilled Qwen3-14B consistently benefit from our method. (`Figure 5`)
>     - **Observation compression reduces both peak tokens and total API cost** as it preserves KV caching, effectively complementing history compression. In AppWorld, it reduces total API cost by about 15% when using the Qwen3-14B compressor (`Figure 7`).
>
> We have already acknowledged this limitation directly in the paper, and the contributions for the proposed problem remain valid. We also acknowledge the reviewer’s concern regarding API cost. While our primary focus is token efficiency and memory cost, we have expanded the discussion of practical API cost in the revision. `Section 4.5` (Cost analysis with an API cost proxy) now provides a more detailed examination of the API cost of our method.
>
> [1] Zhang et al., LightThinker: Thinking Step-by-Step Compression, EMNLP 2025
>
> [2] Zhou et al., MEM1: Learning to Synergize Memory and Reasoning for Efficient Long-Horizon Agents, preprint

---

> ### Author Response · Authors · 2025-11-19
> **Response to Reviewer PhCp (2/3)**
>
> > W2. Cost and Scalability of the Optimization Process Itself: The paper details the effectiveness of the ACON framework but does not sufficiently discuss the "meta-cost" of the guideline optimization process. This process requires running multiple full trajectories (both with and without compression) and then using a powerful "optimizer" LLM for analysis. For new, complex domains, this optimization phase could be prohibitively expensive and time-consuming. The scalability of this approach to a wide variety of new tasks without incurring substantial upfront costs is unclear.
>
> Thank you for raising this important point. We would like to clarify that **the guideline optimization cost is not prohibitively expensive or time-consuming.** It requires fewer than 100 examples, only two trajectories per example, and costs about $22 for an AppWorld scale domain. The resulting guidelines are reusable across tasks, making the approach practical and scalable.
> - The optimization uses at most two trajectories per training example (with and without compression) and fewer than 100 examples per domain (`Appendix B.1`). This cost is far lower than RL methods such as GRPO [1], which require several trajectories for advantage estimation.
> - For AppWorld, **the total cost for optimization steps with o3 is only $2**. The gpt-4.1 rollout cost with and without compressions on 90 examples costs about $19, which corresponds simply to executing the tasks.
> - Rollout and optimization are performed once per domain, and the **resulting guideline can be reused across tasks and smaller compressor models**, keeping practical deployment cost low.
>
> We acknowledge our initial submission did not discuss this clearly. We included this discussion in `Section 4.5` (Optimization cost) of the revision, and we are grateful to the reviewer for giving us the opportunity to make this strong point of our work more explicit.
>
> [1] Shao et al., DeepSeekMath: Pushing the Limits of Mathematical Reasoning in Open Language Models
>
> ---
>
> > W3. Generalizability of Optimized Guidelines: The experiments show that guidelines optimized on a specific benchmark's training set work well on its test set. However, the generalizability of these highly specialized guidelines across different domains remains an open question. For instance, would a guideline optimized for AppWorld's application-based tasks be effective for a radically different domain like code generation or scientific literature review? The paper could be strengthened by including an experiment that tests this cross-domain transferability.
>
> We agree that cross-domain generalization of guidelines is interesting, but **our experiments show that environment-specific guidelines are necessary for decent performance**. This is why ACON includes an automatic guideline optimization pipeline rather than relying on a universal prompt.
> - The naive LLM prompting baseline uses an environment-agnostic prompt and performs noticeably worse, showing that a single universal guideline is insufficient.
> - Our method therefore learns environment-specific rules, which leads to strong performance while avoiding manual prompt engineering. At the same time, **the optimization process itself is domain-agnostic**, meaning it can automatically derive an optimized guideline from training examples in any domain.
> - **Since optimization costs are low as in W2 answer, obtaining guidelines per domain is practical.**
> - Identifying a domain-agnostic compression guideline is an interesting direction, but it is beyond the scope of the environment-specific compression problem we address.

---

> ### Author Response · Authors · 2025-11-19
> **Response to Reviewer PhCp (3/3)**
>
> > Q1. Regarding the critical issue of computational overhead: Could the authors provide a more detailed analysis of the trade-off between peak token reduction and total API cost, especially for history compression? For what types of tasks or interaction patterns does the cost-saving from reduced context outweigh the extra cost of the compressor calls and KV-cache invalidation?
>
> **The trade-off depends on the compression frequency and the compression threshold affects frequency.** Smaller thresholds reduce peak tokens more aggressively but increase compressor calls and KV-cache invalidations. Conservative thresholds reduce both peak tokens and total API cost. History compression is most beneficial when the context grows slowly and the compressed history can be reused across many turns.
> - Smaller thresholds trigger compression more frequently, which lowers peak tokens but increases total API cost due to repeated compressor calls and KV cache disruption (`Figure 6`).
> - In our setting (gpt-4.1, Appworld), we observe the following pattern:
> | Threshold | Peak tokens ($10^3$) | Agent API cost ($) | Compressor API cost ($) |
> |----------:|------------:|---------------:|--------------------:|
> | 2,048     | 5.36        | 0.1440         | 0.0720              |
> | 4,096     | 7.33        | 0.1253         | 0.0360              |
> | 8,192     | 8.70        | 0.1179         | 0.0140              |
> - Aggressive compression greatly reduces peak memory but may increase total cost. More conservative thresholds reduce both compressor usage and total API cost.
> - In practice, history compression is most helpful when context grows slowly and the compressed history can be reused across many turns, so the one time compression cost is amortized over multiple interactions.
>
> We included this discussion and clarified threshold behavior in `Section 4.5` (Cost analysis with an API cost proxy) of the revision.
>
> ---
>
> > Q2. Could the authors elaborate on the cost and complexity of the guideline optimization phase? What is the estimated computational cost (e.g., in terms of number of LLM calls or GPU hours) required to generate an optimized guideline for a new benchmark of similar complexity to AppWorld?
>
> **The guideline optimization phase is lightweight.** For an AppWorld-scale domain, it requires fewer than 100 examples, two trajectories per example, and costs $21 in total. No GPU hours or model training are needed for the guideline optimization phase.
>
> - Our typical setup uses fewer than 100 training examples and two trajectories per example (with and without compression), plus five candidate prompts per optimization step.
> - The total cost for obtaining an optimized guideline for a new domain is therefore roughly $21.
>     - With gpt-4.1 as the agent, rollout costs are about 10 USD without compression and 9 USD with compression.
>     - **The o3-based optimization phase costs about $2.**
> - **The process requires no GPU training and can be run once per domain.**
>
> We included this cost estimate and breakdown more explicitly in `Section 4.5` (Optimization cost) of the revision.
>
> ---
>
> > Q3. The quality of the optimized guideline seems to depend heavily on the capability of the "optimizer" LLM (O3 model in this case). How sensitive is the final guideline quality to the choice of this optimizer? If a less capable model (e.g., gpt-4.1-mini) were used as the optimizer, would the process still yield significant improvements, or does ACON fundamentally rely on having access to a state-of-the-art reasoning model for optimization?
>
> ACON benefits from a reasonably strong LLM optimizer, **but it does not rely on the single best model.** Optimized guideline from gpt-4.1 already gives clear gains over the naive baseline, and o3 provides only additional improvement at low extra cost.
>
> - In `Table 3`, **using gpt-4.1 as the optimizer (47.6) already yields significant improvements over naive prompting (43.5)** on AppWorld with gpt-4.1 agent.
> - Using o3 leads to slightly better guidelines, but the gap between gpt-4.1 and o3 (3.6) is smaller than the gap between naive prompting and either optimized version (4.1).
> - **The cost of using o3 in our setting ($2) is significantly smaller** compared to the rollout cost (\$19), so relying on a capable optimizer is not a major practical barrier.

---

> ### Author Response · Authors · 2025-11-28
> **Gentle Reminder for Discussion**
>
> Dear Reviewer PhCp,
>
> We sincerely appreciate your thoughtful and constructive review. Your feedback was very helpful, and we hope that our rebuttal and revision have addressed the concerns you raised. To summarize the key clarifications provided during the rebuttal:
> - We expanded the **discussion on practical cost and clarified the behavior of API cost, history compression, and peak token reduction**. We included a clearer analysis with concrete measurements in `Cost analysis with an API cost proxy` paragraph in `Section 4.5` of the revision.
> - We clarified the **optimization cost** of the guideline and showed that **it is lightweight**. It requires fewer than 100 examples, only two trajectories per example, and costs about $20 for an AppWorld scale domain. We added an `optimization cost` paragraph in `Section 4.5` of the revision.
> - We clarified why environment specific guidelines are necessary for strong performance. **The optimization procedure is domain-agnostic** and can be reused across tasks, making it practical even when guidelines differ by domain.
>
> We truly appreciate the time and insight you dedicated to reviewing our work. If any of your concerns remain insufficiently addressed, please let us know. We are more than happy to discuss them in further detail.
>
> Best regards,
>
> The Authors

---

### Official Review · Reviewer_AGCY · 2025-10-18

**Soundness:** 3
**Presentation:** 3
**Contribution:** 3
**Rating:** 6
**Confidence:** 4

**Summary:**

Large language models serve as agents in dynamic environments where they accumulate extensive interaction histories, leading to increased computational costs and inefficiencies in long-horizon tasks. The motivation arises from the need to compress these growing contexts effectively, as prior methods primarily address single-step or domain-specific scenarios and fail to preserve essential multi-step signals. Challenges involve retaining diverse information such as states, causal relations, preconditions, and decision cues across heterogeneous tools without losing critical details. ACON introduces a unified framework that optimizes compression guidelines through natural language failure analysis and distills them into smaller models to achieve efficient, informative condensations.

**Strengths:**

1. ACON reduces peak tokens by 26-54% while preserving or enhancing task performance. This efficiency stems from targeted compression that eliminates redundancies without sacrificing key information. Agents can thus handle longer horizons more cost-effectively.

2. The guideline optimization leverages contrastive feedback from successful and failed trajectories. This process refines prompts in natural language space to better capture task-specific needs. As a result, compression becomes more adaptive and effective across diverse environments.

3. Experiments demonstrate consistent gains on AppWorld, OfficeBench, and Multi-objective QA benchmarks. These validations cover varied domains like productivity and question answering. The broad applicability underscores the framework's robustness.

4. ACON improves smaller agents' performance by 20-46% by mitigating long-context distractions. Concise summaries focus reasoning on essential details. This equalization empowers less capable models to tackle complex tasks.

5. The method operates gradient-free, making it suitable for API-based LLMs. No parameter updates are required during optimization. This flexibility supports integration with proprietary systems.

**Weaknesses:**

1. The optimization phase demands collecting feedback from multiple trajectories. This requires significant upfront computation for contrastive pairs. Deployment in time-sensitive scenarios becomes challenging.

2. Benchmarks are simulated and may not capture real-world variability. Unforeseen environmental changes could degrade performance. Broader testing in live settings is essential.

3. Distillation incurs a minor performance drop despite high retention. Critical applications risk failures from lost nuances. Enhanced techniques to minimize this gap are necessary.

4. Thresholds for invoking compression need per-benchmark tuning. Suboptimal values lead to either excessive calls or insufficient reduction. This hyperparameter dependency complicates usage.

5. Comparisons omit some recent agent-specific compression methods. Relative advantages remain unclear without these baselines. Expanding evaluations could better position ACON.

**Questions:**

See Weaknesses.

---

> ### Author Response · Authors · 2025-11-19
> **Response to Reviewer AGCY (1/2)**
>
> We sincerely thank the reviewer for their valuable and insightful feedback.
> We appreciate that the reviewer recognized that our work (1) enables agents to operate over longer horizons more cost effectively, (2) provides a more adaptive and effective compression method across diverse environments, (3) demonstrates broad applicability with consistent gains on three benchmarks, and (4) improves the performance of smaller agents.
>
> To address your concerns,
> - We provide concrete measurements of the optimization cost.
> - We clarify benchmarks used.
> - We explain how the distillation preserves performance with small compressors.
> - We clarify the hyperparameter sensitivity of threshold choices.
> - We add a clearer comparison to recent methods such as MEM1.
>
> Detailed responses follow below.
>
> ---
>
> > W1. The optimization phase demands collecting feedback from multiple trajectories. This requires significant upfront computation for contrastive pairs. Deployment in time-sensitive scenarios becomes challenging.
>
> We appreciate your thoughtful comments on optimization costs. **We would like to clarify that the optimization cost is small and one time.** It uses fewer than 100 examples and costs about $20 on AppWorld. Once a guideline is optimized, it is reused without additional computation.
> - **Our optimization works with fewer than 100 training examples** (`Appendix B.1`). For AppWorld, collecting contrastive trajectories with gpt-4.1 costs approximately **$20 in total**.
> - In many realistic online agent settings, rollout cost overlaps with normal task execution. As a result, the extra cost for optimization becomes even smaller and mainly comes from the o3 based prompt learning steps, which cost about $2.
> - **The optimization is performed only once per environment.** Afterward, the guideline is reused for tasks in the same environment during deployment without any further optimization or additional rollouts.
>
> We clarified this point in `Section 4.5` (Optimization cost) of the revision to make the optimization cost and its one time nature clear to readers.
>
> ---
>
> > W2. Benchmarks are simulated and may not capture real-world variability. Unforeseen environmental changes could degrade performance. Broader testing in live settings is essential.
>
> We agree that real world testing is important. **Our benchmarks, although simulated, including diverse tasks, databases, and documents with clear train-test shifts**, which already captures meaningful variability for evaluating robustness.
>
> - The benchmarks span heterogeneous settings and differing train and test task distributions, with AppWorld in particular offering a highly diverse underlying database of 101 tables, realistic digital activity from about 100 fictitious users, and 750 tasks covering multi-app workflows. This diversity allows us to isolate compression effects under realistic variability.
>
> We also noted real-world evaluation as an important direction for future work in `Appendix A` of the revision.
>
> ---
>
> > W3.  Distillation incurs a minor performance drop despite high retention. Critical applications risk failures from lost nuances. Enhanced techniques to minimize this gap are necessary.
>
> We agree that small drops may matter in sensitive settings. **Our contribution is to show that distillation enables much cheaper models, such as Qwen3-14B, to achieve near teacher level performance** (about 95% of gpt 4.1 in `Figure 4` AppWorld).
> - This preserves the ability to perform effective compression without relying on a large and expensive teacher model, making small compressors far more practical than before.
> - We agree that the remaining gap can be further reduced. More training examples could help, although agent datasets typically contain fewer than 100 training examples.
>
> We included this as a meaningful direction for future refinement in `Appendix A` of the revision.
>
> ---
>
> > W4. Thresholds for invoking compression need per-benchmark tuning. Suboptimal values lead to either excessive calls or insufficient reduction. This hyperparameter dependency complicates usage.
>
> We agree that threshold choice affects performance. This dependency is inherent to compression based long-horizon agents, since the timing of compression directly controls how much context is preserved.
> - Our aim is not to remove this dependency but to make it easier to understand and control.
> - `Figure 6` provides ablations showing how performance varies across thresholds, helping practitioners identify effective operating ranges.

---

> ### Author Response · Authors · 2025-11-19
> **Response to Reviewer AGCY (2/2)**
>
> > W5. Comparisons omit some recent agent-specific compression methods. Relative advantages remain unclear without these baselines. Expanding evaluations could better position ACON.
>
> Thank you for suggesting the comparison with recent agent-specific compression methods. We would like to clearly discriminate ours with MEM1 [1] as follows:
>
> | Aspect | ACON (ours) | MEM1 |
> |--------|-------------|------|
> | **Training requirement** | No model training required | Requires RL training on the agent model |
> | **Applicability to proprietary models** | Applicable (e.g., gpt-4.1) | Not applicable (needs model gradients) |
> | **Agent and compressor design** | Decoupled: large agent + small compressor possible | Coupled: agent and compressor must be the same model |
> | **Objective** | Guideline based controllable compression | Learned compression policy via RL |
>
> - MEM1 adopts a reinforcement learning based compression policy that requires the agent model to be fully trainable. This makes the method incompatible with proprietary models such as gpt-4.1 and forces the agent and compressor to be the same model.
> - In contrast, ACON does not require any model training and supports decoupled architectures where the agent can be large and the compressor can be a small model. Therefore, the design goals and applicable settings differ substantially, which makes direct comparison less straightforward.
>
> We included this comparison in `Appendix C` (Comparison with MEM1) of the revision.
>
> [1] Zhou et al., MEM1: Learning to Synergize Memory and Reasoning for Efficient Long-Horizon Agents, preprint

---

> ### Author Response · Authors · 2025-11-28
> **Gentle Reminder for Discussion**
>
> Dear Reviewer AGCY,
>
> We sincerely appreciate your time and consideration. Your positive review has been encouraging. We hope that our rebuttal and revision have addressed the concerns you raised. To summarize the key clarifications provided in detail during the rebuttal:
> - We clarified the **optimization costs** in detail, showing that the cost is feasible at around 20 dollars per benchmark. We added an `optimization cost` paragraph in `Section 4.5` of the revision.
> - We included a detailed **comparison with the recent agent-specific compression method**, MEM1. We added a `Comparison to Mem1` paragraph in `Appendix C`.
> - We also expanded the explanations regarding the benchmarks used, the distilled compressor performance, and the thresholds for invoking compression.
>
> Your feedback has been very helpful in improving the clarity and completeness of the work. If you have any remaining questions or if further clarification would be useful, we would be happy to discuss them.
>
> Best regards,
>
> The Authors

---

### Official Review · Reviewer_RzY6 · 2025-10-30

**Soundness:** 1
**Presentation:** 3
**Contribution:** 2
**Rating:** 2
**Confidence:** 3

**Summary:**

This authors proposed a framework called ACON designed to reduce the computational cost of LLM agents for long-horizon tasks. The authors identify the growing context length due to accumulated histories of actions and observations as a key obstacle to efficiency. ACON tackles this by introducing a compression guideline optimization that learns how to summarize and retain essential information across the steps in long horizon tasks through a contrastive, min–max formulation. The authors also experimented with distillation of the learned compressor into smaller models using LoRA fine-tuning. Experiments on three benchmarks of AppWorld, OfficeBench, and multi-objective QA show improvements in task success rates and moderate reductions in peak input tokens. However, while ACON achieves better reasoning stability, the actual efficiency gains in terms of total token usage and runtime cost are less convincing.

**Strengths:**

* Formulation of context compression as learning problem: ACON elegantly formulates context compression as a contrastive optimization problem. By pairing successful trajectories with failed ones after compression, it directly trains the model to preserve information that determines success. This min–max objective formalizes what to keep and what to drop in a principled way, moving beyond rule-based or heuristic memory truncation.
* Clear and rigorous methodological description: The paper provides a detailed explanation of the compression guideline optimization process. The use of LLM-as-a-judge evaluation for multiple candidate guidelines, iterative feedback generation, and adaptive prompt selection is described with strong clarity. This makes the method reproducible and highlights the thoughtfulness of the design.
* Exhaustive evaluations with multiple benchmarks: The authors conducted experiments on three distinct benchmarks under varying conditions and provided detailed analyses to understand various aspects of the proposed framework. Evaluations on three distinct long-horizon agentic benchmarks demonstrate consistent improvements in accuracy and moderate token reductions. The inclusion of both full-scale and distilled compressors supports the framework’s flexibility and practical deployment value.
* Strong contribution to reasoning stability: Even though ACON’s original goal was efficiency, its most significant contribution appears in reasoning stabilization. Compressed and structured contexts improve coherence in long-horizon planning, reducing redundant exploration and logical drift in LLM agents.

**Weaknesses:**

* Limited resolution of the claimed efficiency problem: Although the paper motivates ACON as a solution to computational inefficiency caused by long contexts, experiments show that overall token usage and runtime cost did not decrease significantly. In fact, repeated compressor invocations increased API calls, and the authors acknowledge that execution latency rose. The framework thus enhances task performance but not genuine efficiency
* High optimization cost in the guideline learning phase: The compression guideline optimization is extremely expensive, involving iterative LLM calls across the full D_cont dataset. With 20–25 candidate prompts per iteration and multiple iterations, the process may require hundreds of thousands of API calls and many hours of training time. The paper admits this cost but omits quantitative measurements, treating it as an offline overhead. This weakens the practicality of ACON for large-scale or domain-adaptive deployment. The authors argue that guideline optimization is performed once per domain and reused across tasks. However, in realistic multi-domain settings, new environments would require data collection and retraining that reintroduces the same heavy computational overhead. This limits ACON’s scalability and adaptability for general-purpose agent systems.
* Distillation effect Is limited: The distillation step offers only marginal performance gains. The paper itself notes that even GPT-4.1-mini without distillation performs comparably to distilled small models. Hence, the true utility of distillation lies in cost reduction rather than learning transfer, making its contribution modest.
* Guideline optimization depends heavily on heuristic search: Although the optimization is presented as learning-driven, it fundamentally relies on prompt-based heuristic exploration with LLM-as-a-judge feedback. This process lacks theoretical guarantees of convergence or optimality and may depend heavily on model biases and dataset idiosyncrasies.

**Questions:**

Please refer to the Weaknesses section to address the raised issues.

---

> ### Author Response · Authors · 2025-11-19
> **Response to Reviewer RzY6 (1/3)**
>
> We thank the reviewer for their valuable and thoughtful feedback.
> We are glad to hear that you found our work (1) offers an elegant formulation of context compression as a learning problem, (2) provides a clear and reproducible methodology, (3) includes extensive evaluations across multiple benchmarks, and (4) contributes meaningfully to improving reasoning stability.
>
> To address your concerns,
> - We clarify that the scope of our efficiency claim is in memory efficiency.
> - We add quantitative measurements for the guideline optimization cost and suggest its cost is practically low (e.g., $2 for optimization using o3).
> - We emphasized the intended role of distillation in reducing compressor cost.
> - We further explained the motivation behind our optimization procedure.
>
> Details are provided below.
>
> ---
>
> > W1. Limited resolution of the claimed efficiency problem.
>
> Thank you for pointing out this important point. We acknowledge our prior definition of “efficiency” and “cost” was unclear. We would like to emphasize that **ACON mainly targets context explosion problem**, while keeping the total inference/API cost low. Our research direction focuses on reducing **memory cost** (e.g., peak tokens and dependency (accumulated token usage) as used in prior works [1,2]) while preserving performance, unlike baselines.
> - **Our primary objective is to prevent context explosion**, which increases memory usage and degrades the performance of smaller agents.  As stated in the `Section 1` and `Figure 2`, ACON is designed to reduce accumulated context while preserving performance without significantly increasing the inference API cost. **Addressing accumulated context problems has practical impact as it lowers memory cost and improves the stability of small agents** as shown in `Table 1`, `Table 2`, and `Figure 5`.
> - As shown transparently in `Figure 7` and `Section 4.5`, the history compression can result in no gain on API cost reduction. We acknowledge this limitation while emphasizing that **our contributions on the proposed problems remain valid.**
> - Furthermore, **API cost increment is minimal and depends on task characteristics**: in AppWorld, history compression increases cost by 4% while **it decreases cost by 4% in 8-objective QA**. The revised paper (`L500-526`) provides additional analysis.
>
> We also acknowledge that some phrasing in the original submission may have suggested that our primary goal was improving overall efficiency. To avoid this confusion, we revised the terminology in the revision. For example, we replaced cost with memory cost and efficiency with token efficiency so that the contribution is communicated accurately.
>
> [1] Zhang et al., LightThinker: Thinking Step-by-Step Compression, EMNLP 2025
>
> [2] Zhou et al., MEM1: Learning to Synergize Memory and Reasoning for Efficient Long-Horizon Agents, preprint

---

> ### Author Response · Authors · 2025-11-19
> **Response to Reviewer RzY6 (2/3)**
>
> > W2. High optimization cost in the guideline learning phase
>
> We appreciate the reviewer taking the most important and missing analysis for our method. **We would like to clarify that our optimization cost is not significantly high, but scalable and adaptable.** In practice, the guideline optimization costs **less than $2** per benchmark with the o3 model. Rollout cost with and without the compression is less than $20, and this can be adjusted based on budget by controlling the number of training examples or by treating rollout cost as part of normal online task execution.
>
> - For all benchmarks in the main paper, optimization uses one UT step and one CO step (`L1053-1068`), five candidate prompts per iteration (`L1054`), and fewer than 100 training examples (`Section B.1`). This is **much smaller** than the reviewer’s estimate.
> - Concrete cost breakdown for Appworld (90 examples):
>     1. Optimization cost with o3: $1.9
>         1. Utilization maximization (UT) step: $1.2
>         2. Compression maximization (CO) ste: $0.7
>     2. Rollout cost with gpt-4.1: $19.6
>         1. Rollout without compression: $10.7
>         2. Rollout with compression on successful examples from step 2-2: $8.9
>     3. **Total: $21.5**
> - This is much cheaper than reinforcement learning based approaches such as GRPO [1], which require large numbers of rollouts per example and gradient updates on GPUs.
> - Adapting to a new domain requires rerunning rollouts and optimization, but **the total cost remains around $20 for an AppWorld scale domain**, which is small and fully offline. Since rollout collection dominates the cost, users can further reduce cost by using fewer training examples when needed.
> - In many realistic online agent settings, rollout cost overlaps with normal task execution. As a result, the additional cost for optimization is even smaller and mostly comes from the o3 based prompt learning steps.
>
> We included this quantitative cost analysis and breakdown in `Section 4.5` (Optimization cost) of the revision. Your comment significantly improved the clarity of our work. If this concern played a major role in your evaluation, **we would be grateful if you could consider this clarification in your overall assessment.**
>
> [1] Shao et al., DeepSeekMath: Pushing the Limits of Mathematical Reasoning in Open Language Models
>
> ---
>
> > W3. Distillation effect Is limited: The distillation step offers only marginal performance gains.
>
> We appreciate the reviewer’s comment. We agree that the previous caption of `Figure 4` and `Section 4.3` may have been misleading. We would like to clarify that **distillation is not meant to surpass gpt-4.1-mini**. Its purpose is to enable smaller models to replace the large compressor while maintaining effectiveness, which **reduces API cost without sacrificing compression quality.**
>
> - The goal of `Figure 4` is not to show that distilled models outperform gpt-4.1-mini. Rather, **it highlights that small compressor models, including both gpt-4.1-mini and our distilled variants, can perform compression effectively** without requiring the same model size as the agent  (`L412-419`).
> - **The appropriate baseline for interpretation is the large teacher compressor (gpt-4.1), shown by the gray dashed line in `Figure 4`.** Distilled models remain competitive with this teacher despite being much smaller.
> - **The main benefit of distillation is API or computational cost efficiency.** It enables the system to replace a large compressor with a smaller model without losing compression effectiveness. This removes the need to use a large model as both agent and compressor, which substantially lowers API cost. For example, in `Figure 7`, the per example API cost decreases from  $0.164 with gpt-4.1 history compression to 0.124 with Qwen3-14B history compression, which is a 24% reduction.
>
> We clarified this point in `Section 4.3` of the revision.

---

> ### Author Response · Authors · 2025-11-19
> **Response to Reviewer RzY6 (3/3)**
>
> > W4. Guideline optimization depends heavily on heuristic search.
>
> We agree that moving beyond heuristic and model bias sensitive prompt optimization is an important research direction. However, we would like to emphasize that **our contribution is to show that a natural language based optimization is already effective for the context compression problem we study.**
> - In this paper, we demonstrate that a natural language based optimization, despite its known limitations, is **effective in our setting and works directly with black-box models without requiring access to model weights or GPU training.** This makes it a practical, low cost, and broadly applicable alternative to more resource intensive training-based methods.
> - These limitations are shared by existing prompt optimization methods [1,2]. We appreciate the reviewer highlighting this point and clarified it in `Appendix A` of the revision so that readers understand both the strengths and the inherent limitations of this class of approaches.
>
> [1] Khattab et al., DSPy: Compiling Declarative Language Model Calls into Self-Improving Pipelines, ICLR 2024
>
> [2] Yuksekgonul et al., TextGrad: Automatic "Differentiation" via Text, Nature

---

> ### Comment · Reviewer_RzY6 · 2025-11-25
> **Thank you for the clarifying responses**
>
> The responses address most of my concerns especially with the optimization cost, so I am raising my score to 6.

---

> > ### Author Response · Authors · 2025-11-25
> > **Response to Reviewer RzY6**
> >
> > Dear Reviewer RzY6,
> >
> > Thank you for your active engagement, thoughtful consideration, and positive assessment of our work. We truly appreciate your decision to raise the score from **2 to 6**, and we are glad that the **rebuttal addressed most of your concerns**, including those related to optimization cost.
> >
> > Your feedback has been invaluable in improving the clarity and completeness of the paper. If you have any remaining questions or concerns, please let us know. We are happy to discuss them further.
> >
> > Best Regards,
> >
> > The Authors

---

### Author Response · Authors · 2025-11-21
**General Response to Reviewers**

We thank the reviewers for their time, effort, and constructive feedback, which strengthened this work through rebuttal.
Reviewers highlighted that our framework provides an **elegant formulation of context compression as a learning problem** and a clear, reproducible methodology with **extensive evaluations** across multiple benchmarks [RzY6]. They also noted that **ACON enables agents to operate over longer horizons more cost effectively**, offers a more adaptive and effective compression method across diverse environments, demonstrates broad applicability with consistent gains on three benchmarks, and **significantly improves the performance of smaller agents** [AGCY, PhCp]. We are encouraged that reviewers found our framework promising for robust long-horizon agents, especially for smaller agents.

Below we summarize the key clarifications added during the rebuttal:

- **ACON targets memory efficiency rather than overall API cost [RzY6, PhCp]**. We clarified that our main goal is reducing peak tokens and dependency, and reducing accumulated context is practically important even when API costs remain similar. We updated terminology to avoid implying improvements in overall efficiency.
- **Optimization cost is low, scalable, and one-time [RzY6, AGCY, PhCp]**. Our optimization requires fewer than 100 examples and two trajectories per example, which costs about only 20 USD per domain. with o3 optimizer accounting for only 2 USD or less in this process.
- **Distillation enables small compressors to retain strong compression quality [RzY6, AGCY]**. We clarify that distillation is not intended to outperform gpt-4.1-mini; instead, it enables replacing large compressors with substantially cheaper small models while preserving compression effectiveness.

We also added clarifications addressing the specific questions raised by each reviewer.
In addition, we updated the manuscript as follows:
- We added `Optimization cost` paragraph under `Section 4.5` to clarify the concerns regarding the practicality of our framework [RzY6 W2, AGCY W4, and PhCp W2].
- We rewrote the `Limitation: Cost analysis` paragraph into a new `Cost analysis with an API cost proxy` under `Section 4.5` to clarify the framing of our work and provide detailed analysis of API cost trends [RzY6 W1, PhCp W1].
- We added a `Comparison with MEM1` paragraph in `Appendix C` and updated `Section 2` (Related Works) to clarify the distinction between our method and a recent baseline [AGCY W5].
- We updated parts of the paper to address points of misunderstandings. These changes are marked in blue.

We thank the reviewers again for the constructive insights that helped sharpen the scope, clarity, and positioning of this work.

---

### Author Response · Authors · 2025-12-01
**Public Letter to AC/Reviewers**

Dear AC and Reviewers,

We would like to express our sincere gratitude to the reviewers for their constructive feedback and time for reviewing our work. We also deeply appreciate the new Area Chair for stepping in to handle this challenging situation.

To assist your evaluation, we would like to provide a summary of our rebuttal progress and the consensus that was forming immediately prior to the system reversion.

**Overall Status: Score reached 6/6/4 (improved from 2/6/4).** Before the reviews were reverted, our paper successfully improved its score through the rebuttal process, demonstrating a clear trajectory of resolving concerns and building a positive consensus as follows:

1. **Reviewer RzY6 (Initial: 2 $\rightarrow$ Final: 6):** On **November 25, 2025** (two days prior to the official incident notification on Nov 27), Reviewer RzY6 raised their score from 2 to 6.
    - Reviewer RzY6 explicitly stated that our rebuttal responses successfully addressed most of their concerns, especially regarding the optimization cost.
    - This timeline confirms that the score increase was a result of legitimate scientific discussion, unrelated to the recent irregularities.
2. **Reviewer PhCp (Score: 4):** Due to the halt in discussion, we did not have the opportunity to conclude the discussion with Reviewer PhCp. However, we would like to clarify that the remaining major concerns are effectively addressed by our rebuttal:
    - **Optimization Cost (Effectively Resolved):** This concern (PhCp W2) was shared by Reviewer RzY6 (W2). As noted above, Reviewer RzY6 explicitly acknowledged that our rebuttal demonstrating that the cost is **not expensive** addressed their concern. We believe this serves as strong evidence that PhCp's concern is also effectively resolved by our response.
    - **Generalizability (Addressed by Low Cost & Robustness):** Regarding the concern on cross-domain transferability (PhCp W3), we emphasized that: (1) Given the lightweight optimization cost (validated by RzY6), users can efficiently optimize guidelines for each specific domain; and (2) Our method’s consistent performance across three benchmarks demonstrates that the optimization framework itself is domain-agnostic.
    - **Practical Cost vs. Token Efficiency (Missed Discussion):** Regarding W1, we lost the opportunity to discuss it. We aimed to clarify that our method's significant memory efficiency gains (up to 54%) highly outweigh the marginal practical cost increment (up to 4%). We believe a brief discussion regarding our analysis of this trade-off could have effectively addressed the reviewer's concern.

We respectfully request that the AC consider these points: **the proven validity of our rebuttal (evidenced by RzY6's score change) and the high probability that the remaining concerns were resolvable.**

Once again, thank you for your time and dedication to the review process. We are fully available to answer any questions the new AC may have.

Best Regards,

The Authors

---

### Meta-Review · Area_Chair_Ly4M · 2026-01-05

**Summary:**

### summary

This authors proposed a framework called ACON designed to reduce the computational cost of LLM agents for long-horizon tasks. The authors identify the growing context length due to accumulated histories of actions and observations as a key obstacle to efficiency. ACON tackles this by introducing a compression guideline optimization that learns how to summarize and retain essential information across the steps in long horizon tasks through a contrastive, min–max formulation.

### reviewer summary

AGCY highlights per-benchmark tuning burden and missing comparisons to more recent compression methods.
RzY6 moving from 2 to 6 suggests the rebuttal effectively addressed the meta-cost uncertainty, but the initial concerns about the efficiency claim and the heuristic nature of prompt-style search are not truly eliminated.
PhCp highlights the clarity and the thoroughness; core concerns lie in the trade-off between peak token reduction and total API cost, as well as optimizing the meta-cost of guidelines, cross-domain generalization, and optimizer model dependencies.

### AC comments

I agree the paper has a promising and reasonably well-structured approach to optimizing compression guidelines, and the evaluation effort is substantial. However, the central story is not fully closed:

1. The paper motivates itself around efficiency/cost, the empirical story does not consistently show reductions in total token usage / runtime / API cost, and can even worsen because the compressor is invoked repeatedly. In rebuttal, the authors effectively narrow the target to memory/peak-token efficiency and acknowledge that history compression may not reduce API cost. That clarification is good, but it also makes the work feel like it retreats from a broader promise, which tends to hurt in tight competition.

2. PhCp explicitly points out the real-world mismatch: peak tokens can drop while total cost increases, because compression adds extra model calls and can undermine KV caching, leading to repeated compute. The rebuttal adds threshold-based trade-off analysis and shows the cost increase may be small or even slightly negative in some settings, but the overall message becomes it depends on tuning and workload.

3. PhCp also inquired about cross-domain generalization: whether the optimized guidelines could be applied across domains. The authors' response essentially stated, “Environment-specific guidelines are indeed necessary, which is why we provide an automated optimization pipeline; cross-domain guidelines fall outside the scope of this paper.” While logically consistent, this implies the method's deployment model resembles running the process anew for each environment, thereby diminishing its appeal as a “universal approach.”

**Reviewer Concerns:**

Largely addressed:

* Clearer positioning toward memory/peak-token efficiency and concrete cost breakdown claims.
* Clarified the role of distillation as enabling cheaper compressors rather than outperforming strong baselines.


Still outstanding:

* The practical trade-off between peak-token reduction and total cost/runtime remains a major adoption blocker.
* Cross-domain generalization and multi-domain scalability remain unclear without direct evidence.
* Reliability and reproducibility concerns due to heuristic search + LLM-as-judge dependence are not fully resolved.

**Reviewer Scores:**

* RzY6 likely stays at 6.
* AGCY likely stays at 6.
* PhCp would maintain 4, but core concerns on total cost/scalability/generalization.

---

### Decision · Program_Chairs · 2026-01-26

Reject